# Target binding triggers hierarchical phosphorylation of human Argonaute-2 to promote target release

**Brianna Bibel[1,2], Elad Elkayam[2†], Steve Silletti[3], Elizabeth A Komives[3], Leemor Joshua-Tor[1,2]***

[1]Cold Spring Harbor Laboratory School of Biological Sciences, Cold Spring Harbor, United States; [2]Howard Hughes Medical Institute, W. M. Keck Structural Biology Laboratory, Cold Spring Harbor, United States; [3]Department of Chemistry and Biochemistry, University of California, San Diego, La Jolla, United States

**Abstract** Argonaute (Ago) proteins play a central role in post-transcriptional gene regulation through RNA interference (RNAi). Agos bind small RNAs (sRNAs) including small interfering RNAs (siRNAs) and microRNAs (miRNAs) to form the functional core of the RNA-induced silencing complex (RISC). The sRNA is used as a guide to target mRNAs containing either partially or fully complementary sequences, ultimately leading to downregulation of the corresponding proteins. It was previously shown that the kinase CK1α phosphorylates a cluster of residues in the eukaryotic insertion (EI) of Ago, leading to the alleviation of miRNA-mediated repression through an undetermined mechanism. We show that binding of miRNA-loaded human Ago2 to target RNA with complementarity to the seed and 3' supplementary regions of the miRNA primes the EI for hierarchical phosphorylation by CK1α. The added negative charges electrostatically promote target release, freeing Ago to seek out additional targets once it is dephosphorylated. The high conservation of potential phosphosites in the EI suggests that such a regulatory strategy may be a shared mechanism for regulating miRNA-mediated repression.

*For correspondence:
leemor@cshl.edu

Present address: †Ventus Therapeutics, Waltham, Massachusetts, United States

## Editor's evaluation

This paper provides well documented and solid biochemical data to show how phosphorylation of hAGO2 modulates RISC-target mRNA binding dynamics. The results explain how hAGO2 is released from a target allowing a limited pool of Ago proteins to target a very large repertoire of mRNA molecules. The new data and discussion support the key claims of the manuscript.

## Introduction

Most metazoan genes are regulated post-transcriptionally by RNA interference (RNAi) (*Friedman et al., 2009*). RNAi is a conserved process whereby an Argonaute (Ago) protein binds to a small RNA, such as a microRNA (miRNA) or a small interfering RNA (siRNA), ~22 nt in length, and uses it as a guide to seek out and bind target mRNAs containing regions of partial complementarity (*Friedman et al., 2009*; *Bartel, 2018*). Binding of the core RNA-induced silencing complex (RISC), consisting of Ago and a guide (*Rivas et al., 2005*), to an mRNA can lead to recruitment of cofactors and decreased translation of protein from that mRNA through a number of mechanisms including: preventing translation initiation and elongation; promoting mRNA decay; triggering premature termination and co-translational protein degradation; and sequestering targets from the translational machinery (*Jonas and Izaurralde, 2015*; *Eichhorn et al., 2014*). Ago has a high affinity for miRNAs with a very slow off-rate

**eLife digest** Proteins are the chemical 'workhorses' of the cell: some help maintain a cell's shape or structure, while others carry out the chemical reactions necessary for life. Organisms therefore need to keep tight control over the production of proteins in their cells, so that the right amount of each protein is made at the right time, in the right place.

Instructions for making new proteins are encoded in a type of molecule called messenger RNA. Each messenger RNA contains the instructions for one protein, which are then 'read' and carried out by special cellular machinery called ribosomes. The cell can control how much protein it produces by regulating both the levels of different messenger RNA and the amount of protein ribosomes are allowed to make from those instructions.

The main way to regulate the levels of messenger RNA is through their transcription from the genome. However, this needs fine tuning. Cells can do this in a highly specific way using molecules called microRNAs. A microRNA works by directing a protein called Argonaute to the messenger RNA that it targets. Once Argonaute arrives, it can call in additional 'helper proteins' to shut down, or reduce, protein production from that messenger RNA, or alternatively to break down the messenger RNA altogether.

Cells can use an enzyme called CK1α to attach bulky chemical groups onto a specific part of the Argonaute protein, in a reaction termed phosphorylation. The ability to carry out this reaction (and to reverse it) also seems to be important for microRNAs to do their job properly, but why has remained unknown. Bibel et al. wanted to determine what triggers CK1α to phosphorylate Argonaute, and how this affects interactions between microRNAs, Argonaute and their target messenger RNAs.

A series of 'test tube' experiments looked at the interaction between purified CK1α and Argonaute under different conditions. These demonstrated that CK1α could only carry out its phosphorylation reaction when Argonaute was already interacting with a microRNA and its corresponding messenger RNA. Further measurements revealed that phosphorylation of Argonaute made it detach from the messenger RNA more quickly. This suggests that phosphorylation might be a way to let Argonaute seek out new messenger RNAs after blocking protein production at its first 'target'.

These results shed new light on a fundamental mechanism that cells use to control protein production. Bibel et al. propose that this mechanism may be shared across many different species and could one day help guide the development of new medical therapies based on microRNAs.

---

(**De et al., 2013**; **Kingston and Bartel, 2019**). Consequently, most Ago/guide complexes are long-lived, with average half-lives on the order of days (**Gantier et al., 2011**) and some complexes stable for at least 3 weeks (**Olejniczak et al., 2013**). Thus, once loaded with a guide, Ago can repress countless matching mRNA sites, as long as it is able to efficiently release these targets once silencing is achieved. This turnover is crucial because potential target sites are present in large excess compared to the corresponding miRNAs (**Denzler et al., 2014**; **Denzler et al., 2016**). miRNA-binding sites are typically located within the target mRNA's 3' UTR and are primarily determined by perfect complementarity in a 6–8 nt 'seed sequence' corresponding to nucleotides 2–8 of the guide miRNA (g2–8) (**Bartel, 2009**), augmented by additional pairing beyond this region. In the core RISC, the subseed region of the miRNA (g2–5) is preorganized for base pairing (**Elkayam et al., 2012**; **Schirle and MacRae, 2012**; **Klum et al., 2018**), which allows Ago to rapidly sample potential target sites (**Klum et al., 2018**). Binding to a fully complementary seed induces conformational changes in Ago that expose more of the miRNA and widen Ago's RNA-binding channel, allowing for further pairing (**Schirle et al., 2014**). Target sites vary in their amount of complementarity to a guide outside of the seed sequence, and additional pairing can influence a site's repression efficacy (**Grimson et al., 2007**). This pairing often occurs in the more solvent-exposed 3' supplementary region (corresponding to g12–17) through a second nucleation event (**Bartel, 2018**; **Sheu-Gruttadauria et al., 2019b**). 3' supplementary pairing, most effectively with 3–4 contiguous base pairs centered around g13–16 (**Grimson et al., 2007**), can strengthen repression by increasing guide/target affinity through a decrease in $K_{off}$ (**Xiao and MacRae, 2020**; **Wee et al., 2012**; **Salomon et al., 2015**). It also allows miRNAs of the same family (miRNAs that share the same seed sequence) to preferentially repress different sets of targets (**Moore et al., 2015**; **Broughton et al., 2016**). More common for siRNAs, full guide-target complementarity is rare among

metazoan miRNA targets, but when present it results in 'slicing' by some Ago proteins, a process in which the target mRNA is cleaved between nucleotides across from g10 to g11 (*Song et al., 2004*; *Liu et al., 2004*; *Meister et al., 2004*).

Ago proteins are part of a larger Ago superfamily, which includes Piwi- and Wago-clade proteins. Members of the superfamily show high structural conservation throughout evolution and have four main structural domains: N, PAZ, MID, and PIWI (*Song et al., 2004*). The MID and PIWI domains form an almost continuous lobe which, together with the N domain, forms a crescent-shaped base above which the PAZ domain is able to move more freely (*Song et al., 2004*; *Ming et al., 2007*). The 5' end of the small RNA is bound by a pocket created by the MID and PIWI domains (*Frank et al., 2010*) and the 3' end is held by the PAZ domain (*Song et al., 2003*; *Ma et al., 2004*), with the length of the small RNA cradled in a groove traversing the crescent (*Elkayam et al., 2012*; *Schirle and MacRae, 2012*; *Wang et al., 2008*). The PAZ domain is the most flexible and, sometimes in concert with the N domain, undergoes rigid body movements along linker regions (L1 and L2) in order to accommodate RNA (*Schirle et al., 2014*; *Sheu-Gruttadauria et al., 2019b*; *Ming et al., 2007*; *Faehnle et al., 2013*). Humans have four Ago orthologs (hAgo1–4) in addition to four Piwi-clade Agos, with hAgo2 being the most highly expressed in most tissues (*Valdmanis et al., 2012*; *Hauptmann et al., 2015*). Although particular functions have been attributed to specific hAgo proteins, hAgo1–4 are largely considered functionally redundant (*Su et al., 2009*). The lone exception is hAgo2; knock-out of hAgo2 is lethal in early mouse development (*Liu et al., 2004*; *Morita et al., 2007*), whereas knock-out of the other hAgos, and even knock-out of all three, is not (*Hajarnis et al., 2018*). This essentiality is attributed largely to the fact that, under most circumstances, only hAgo2 has slicer activity (*Song et al., 2004*; *Liu et al., 2004*; *Meister et al., 2004*). Although slicing is not required for miRNA-mediated regulation, it is necessary for the maturation of rare but important Dicer-independent miRNAs including miR-451 (*Cheloufi et al., 2010*) and miR-486 (*Jee et al., 2018*), which play essential roles in erythrocyte development.

Although the core architecture is conserved among all Ago proteins, they are subject to multiple types of post-translational modification, including ubiquitylation, PARylation, and phosphorylation (*Jee and Lai, 2014*; *Gebert and MacRae, 2019* ). For example, S837 undergoes stress-induced p38/MAPK-mediated S837 phosphorylation, which, among other effects, is thought to contribute to hAgo2 localization in processing bodies (P-bodies) (*Zeng et al., 2008*). There is a span of residues on the surface of the PIWI domain in eukaryotic miRNA-specialized Ago proteins that is absent in prokaryotic Ago proteins (*Figure 1a*). This region, corresponding to residues ~820–837 in hAgo2 numbering, is termed the eukaryotic insertion (EI), and it contains four to five highly conserved potential phosphorylation sites: four serine residues (in hAgo2 numbering, S824, S828, S831, and S834) as well as a threonine (T830) which is only conserved in Ago2 homologs. This cluster was recently found to be heterogeneously phosphorylated in multiple Ago homologs in a variety of cell types and organisms (*Golden et al., 2017*; *Quévillon Huberdeau et al., 2017*) and its phosphorylation and dephosphorylation were attributed to the kinase CK1α and the phosphatase PP6/ANKRD52, respectively (*Golden et al., 2017*).

This cycle of phosphorylation of Ago's EI was implicated in regulatory control of Ago itself (*Golden et al., 2017*; *Quévillon Huberdeau et al., 2017*). Golden et al. found that *decreased* EI phosphorylation led to an increase in target RNA association with the Ago-guide complex in cell culture, as measured by qt-RT-PCR of immunoprecipitated hAgo2, whereas *increased* EI phosphorylation or its mimicking had the opposite effect (*Golden et al., 2017*). Additionally, disruption of the hAgo EI phosphorylation cycle led to dysregulation of mRNA levels, as determined by RNA-seq. In the absence of EI phosphorylation, hAgo2 associated with a more diverse pool of mRNA targets, but with decreased coverage of individual targets as determined by eCLIP (*Golden et al., 2017*). Furthermore, reporter assays showed that increased phosphorylation or its mimicking led to reduced repression of reporter targets in cell culture (*Cheloufi et al., 2010* and in *Caenorhabditis elegans Jee et al., 2018*). These silencing defects could be rescued by tethering Ago to the target, indicating that phosphorylation impaired a step upstream of repression, preventing productive Ago-guide/target interactions (*Golden et al., 2017*; *Quévillon Huberdeau et al., 2017*). Together with their findings that miRNA production and guide association appeared unaffected, these results point to a role of EI phosphorylation in mediating Ago-guide/target interactions, but key questions remained about the molecular events associated with this process.

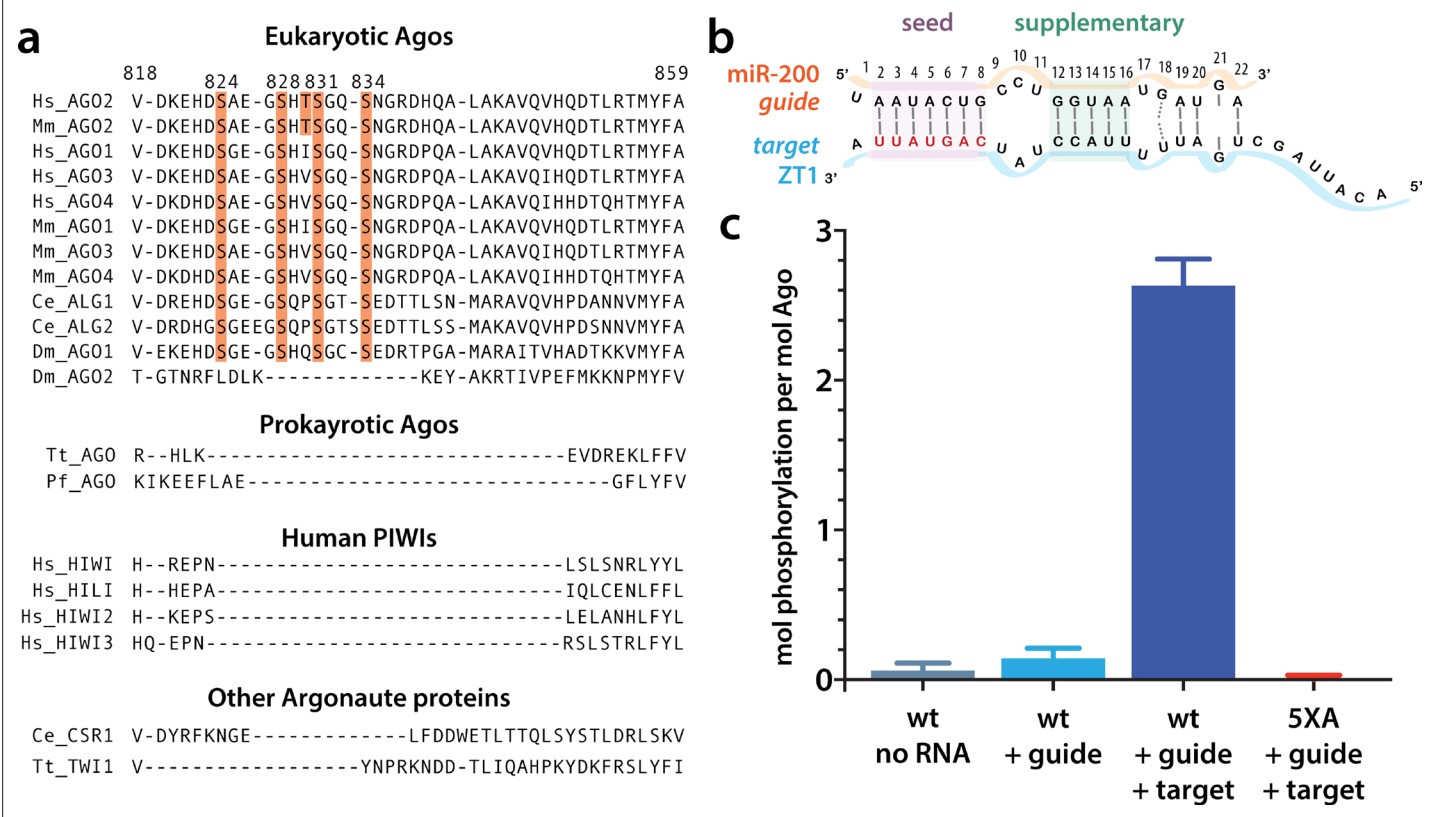

**Figure 1.** Target binding triggers phosphorylation of the hAgo2 eukaryotic insertion (EI). (**a**) Conservation of the EI in eukaryotic miRNA-handling Argonaute proteins. Phosphorylatable residues are highlighted in orange. The four serines are highly conserved in higher eukaryotes, while the threonine is specific to hAgo2. Residue numbering is based on hAgo2. Hs, *Homo sapiens*; Ce, *Caenorhabditis elegans*; Dm, *Drosophila melanogaster*; Mm, *Mus musculus; Pf, Pyrococcus furiosus; Tt, Thermus thermophilus*. (**b**) Guide/target pair schematic for miR-200 and a 30-mer target RNA (ZT1) corresponding to a natural and validated miR-200-binding site located in the 3' UTR of ZEBT1. Nucleotides in the target complementary to the miR-200 seed sequence are labeled in red and the seed and 3' supplementary pairing regions are indicated. (**c**) Target binding, but not guide binding, dramatically increases CK1a-mediated phosphorylation of hAgo2 as measured by in vitro phosphorylation assays using the guide/target pair shown in b. Quantification of phosphorylation was done by liquid scintillation and comparison to a spotted ATP standard. Data are presented as the mean ± standard error (SE) for three technical replicates.

The online version of this article includes the following figure supplement(s) for figure 1:

**Figure supplement 1.** Target binding triggers phosphorylation of the EI in hAgo1 and hAgo3.

Using a combination of biochemical and biophysical assays, we determined that binding of miRNA-bound hAgo2 to targets containing 3' supplementary pairing triggers hierarchical phosphorylation of the EI by CK1α. The added negative charge electrostatically repels the negatively charged target RNA, leading to target release. Such regulation could help explain how cells maintain tight and consistent miRNA-mediated regulation despite there being a large excess of miRNA target sites compared to Ago and miRNA molecules (*Denzler et al., 2014*; *Denzler et al., 2016*).

## Results

### Target binding promotes CK1α-mediated EI phosphorylation in vitro

As noted above, EI is a feature that is characteristic of eukaryotic miRNA-specialized Ago proteins and is absent in prokaryotic Ago proteins. We noticed that it is also absent in siRNA- or piRNA-specialized Ago superfamily proteins (e.g. *Drosophila melanogaster* Ago2 and Piwi proteins, respectively) (*Figure 1a*). Previous work showed that hAgo2 EI phosphorylation was enhanced by the addition of a miRNA accompanied by a miRNA sponge containing multiple binding sites specific to that miRNA (*Golden et al., 2017*). This suggested that, within cells, EI phosphorylation is stimulated by target binding. This stimulatory effect could be due to features inherent to the ternary Ago-miRNA/

target complex itself, or caused by binding of external mediators, for example, repression-mediating proteins that are recruited after target binding such as GW-182, PAN2-PAN3, and CCR4-NOT (*Jonas and Izaurralde, 2015*). To determine if target-binding alone could stimulate phosphorylation in the absence of cofactors, we tested CK1α-mediated phosphorylation of recombinant, dephosphorylated, hAgo2 in its apo state (RNA-free), when bound to a guide miRNA (miR-200b), and when bound to guide and a partially complementary target in vitro. For this target, we used a 30-nt RNA we will refer to as ZT1, which corresponds to a natural, highly conserved, miR-200-binding site in the 3′ UTR of the transcription factor ZEB1 (*Gregory et al., 2008*). This guide/target combination was chosen because the target's genomic location was identified as bound to 5XA but not wild-type (wt) hAgo2 in previous eCLIP experiments (*Golden et al., 2017*), suggesting it was sensitive to EI phosphorylation in vivo. It was also chosen because of its functional relevance in helping regulate epithelial to mesenchymal transitions (*Gregory et al., 2008*). We did not detect any phosphorylation of RNA-free hAgo2 above background and detected very low levels of phosphorylation with guide-only-bound hAgo2, 0.14 ± 0.07 mol phosphorylation per mol Ago2 (*Figure 1*). However, when we added target RNA, we observed substantial phosphorylation, 2.6 ± 0.2 mol phosphorylation per mol hAgo2 (*Figure 1*). No phosphorylation was observed when all five phosphorylatable EI residues were mutated to alanine hAgo2(5XA) (*Figure 1*), confirming that the observed phosphorylation of the wt hAgo2 occurs and is dependent upon EI residues. To test whether the other human Ago orthologs behave in a similar fashion, we tested hAgo1 and hAgo3 using this assay. We found that these Agos are also subject to target-binding-triggered CK1α-mediated phosphorylation in vitro (*Figure 1—figure supplement 1*). This is consistent with previous findings that hAgo1 and hAgo3 are phosphorylated on multiple residues in the EI in vivo (*Quévillon Huberdeau et al., 2017*).

## Phosphorylation is promoted by pairing in the supplementary region

Because of the observed effects of target binding, we decided to further investigate what features of the target RNA affect EI phosphorylation. In order to determine the minimum target length sufficient to trigger it, we performed in vitro phosphorylation assays with a series of ZT1 target RNAs, starting with an 8 nt target complementary to the seed region and extending the 5′ end (corresponding to the 3′ end of the guide) up to a target length of 30 nt. No significant phosphorylation was detectable up to a target length of 13 nt, however, phosphorylation increased dramatically once the target length reached 14 nt, with further increases in phosphorylation observed at longer lengths (*Figure 2a*). These differences could not be explained by differences in binding affinity (*Figure 2—figure supplement 1*).

It is likely that 14 nt represents the shortest length of target capable of establishing stable pairing in the supplementary region. Therefore, to further investigate the importance of supplementary pairing, we tested phosphorylation of hAgo2-guide-traget ternary complexes with varying amounts of complementarity to the guide miRNA. Length-matched (30 nt) target with complementarity to the seed region alone was incapable of stimulating robust hAgo2 phosphorylation (*Figure 2b*). However, targets with full complementarity also failed to trigger phosphorylation in either wt or catalytically inactivated hAgo2 (Ago2(D669N)) (*Liu et al., 2004*). The latter was included to control for potential confounding effects caused by slicing, which occurs with hAgo2 and fully complementary guide/ target pairs. This result is consistent with our finding that EI phosphomimetics do not significantly affect slicing (*Figure 2—figure supplement 2*). Furthermore, targets with extended 3′ complementarity, resembling target-directed miRNA decay (TDMD) targets, which are known to lead to an altered conformation of hAgo2 (*Sheu-Gruttadauria et al., 2019a*), triggered significantly less phosphorylation than the original target (*Figure 2b*). Therefore, length and supplementary region complementarity are not sufficient to maximally trigger phosphorylation. Instead, a more particular target site configuration of seed + supplementary pairing (i.e. without central or extensive 3′ pairing) serves as a trigger.

These findings may be explained by the unique conformations Ago adopts to accommodate varying amounts of base pairing. Binding to short targets is known to lead to a shift in hAgo2's PAZ domain (*Schirle et al., 2014*), which could make the EI more accessible for CK1α. There are no structures of hAgo2 bound to a full-length 'canonical' miRNA target containing seed and supplementary pairing. However, the crystal structure of a catalytically inactivated hAgo2 bound to a target RNA with complementary to nucleotides 2–16 of the guide (g2–16) was recently determined (*Sheu-Gruttadauria et al., 2019b*). In this structure, likely due to Ago's inability to cleave the target, the RNA is not base paired in the central region (g9–11) and instead pairs only in the seed and supplementary regions,

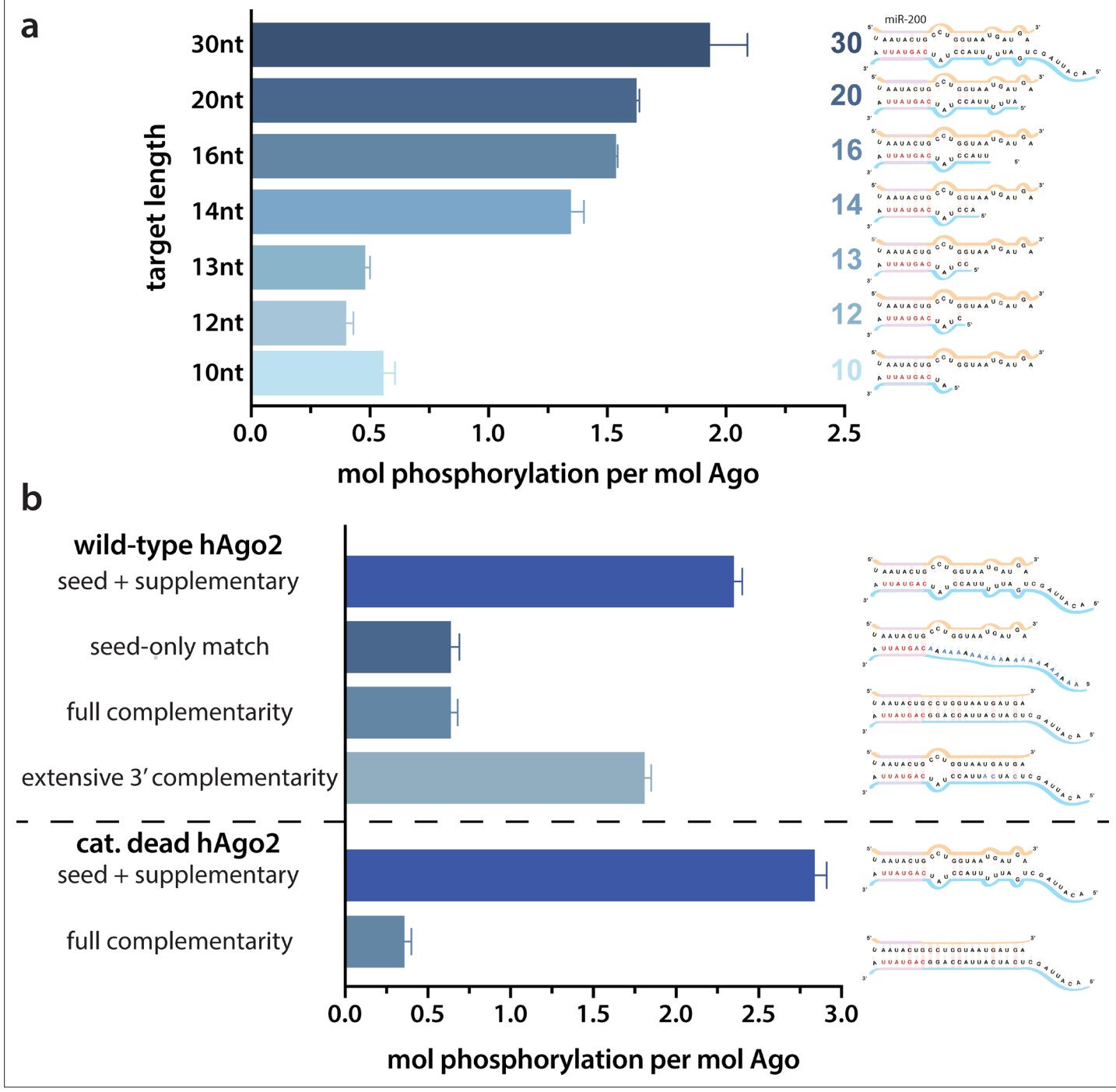

**Figure 2.** Target length and complementarity requirements. (**a**) In vitro phosphorylation assays with ZT1-based targets of varying lengths show that a minimum of 14 nucleotides is needed to trigger robust phosphorylation. (**b**) Assays with ZT1-based targets with different levels of complementarity to the guide (miR-200) show that the seed + 3′ supplementary configuration is best able to induce phosphorylation. A catalytically inactivated (cat-dead) hAgo2, hAgo2(669N), is included to control for possible slicing of the fully complementary target. Quantification of phosphorylation was done by liquid scintillation and comparison to a spotted ATP standard. Data are presented as the mean ± standard error (SE) for three technical replicates.

The online version of this article includes the following figure supplement(s) for figure 2:

**Figure supplement 1.** Strength of target binding to hAgo2 and phosphorylation are not strictly correlated.

**Figure supplement 2.** Phosphomimicking doesn't impact slicing.

a conformation which the authors posit could represent one that is adopted by Ago upon binding traditional miRNAs (*Sheu-Gruttadauria et al., 2019b*). In this structure, the 3' end remains held by the PAZ domain while the N and PAZ domains move away from the MID and PIWI domains, which could make the EI more accessible to CK1α.

## Triggering of phosphorylation upon target binding is not due to major conformational changes in the EI

The EI itself is unresolved in all Ago structures determined to date. In order to obtain structural information regarding conformational changes that the EI might be undergoing during these events, we turned to hydrogen-deuterium exchange mass spectrometry (HDX-MS), which measures conformational dynamics and solvent accessibility by analyzing the exchange of hydrogens for deuterium over time. The measured deuterium uptake corresponds primarily to exchange from backbone amides, and thus regions with stable secondary structure have high levels of protection (low deuterium uptake) whereas more flexible and/or exposed regions are less protected and thus have higher deuterium uptake (*Mandell et al., 1998*; *Wales and Engen, 2006*). Samples of hAgo2 in RNA-free (RF), guide-bound (G-bound), and guide + target-bound (GT-bound) states were analyzed to investigate whether guide- and/or target-binding influence the structure of the EI or the surrounding residues. Since no reliable peptides covering the EI could be measured for wt hAgo2, we used the 5XA mutant, which has a similar target-binding affinity compared to dephosphorylated wt (*Figure 3—figure supplement 1*), yet did have reliable coverage of the EI. The 5XA mutant showed minimal differences in exchange compared to the dephosphorylated wt RNA-free hAgo2 over the rest of the protein (*Figure 3—figure supplement 2*). Very minor, yet statistically significant ($p < 0.01$), differences were only seen at one of the four timepoints (2 min) and were restricted to 10 (out of 255) peptides, none of which were in the region surrounding the EI. Thus, 5XA is a good proxy for the dephosphorylated wt protein. Near-complete (99.4%) sequence coverage was achieved, with a total of 262 peptides and an average redundancy of 5.29 (*Figure 3—figure supplement 3*).

Seven peptides covering the EI were detected in all datasets, the shortest being 811–842 and 812–842. These peptides were consistent with each other, as would be expected given the almost complete overlap between the two. Importantly, they showed no statistically significant differences between the G-bound and the GT-bound at any timepoint ($p < 0.01$) (*Figure 3*). The G-bound showed slight protection compared to the RF, however taking overlapping peptides into account to increase resolution, this increase can be attributed to higher uptake in the region downstream of 834, outside of the EI itself (*Figure 3a*). The EI showed a moderately high level of exchange, suggesting little defined secondary structure in any state, and no detectable differences in secondary structure content between the states. If maximal exchange was already realized at the earliest timepoint (30 s), we would not be able to detect an increase in flexibility nor solvent accessibility, only a decrease. Since we do not see any such decrease, we can conclude that neither guide- nor target-binding offer protection to phosphorylatable residues. However, we cannot rule out conformational changes occurring that do not involve strong changes in secondary structure content.

The lack of differences between the RF and G-bound states within the EI is in stark contrast to the large differences observed outside of the EI. The most substantial protection upon guide-binding occurred in regions known to interact directly with the miRNA, such as the 5' binding pocket (*Figure 4—figure supplement 1*). However, the RF also showed higher relative uptake over most of the protein, suggesting greater overall flexibility prior to guide binding. This is consistent with previous results showing guide binding protects from limited proteolysis (*Elkayam et al., 2012*) and with the observation that RNA-free hAgo2 is unstable in cells and rapidly degraded (*Martinez and Gregory, 2013*), whereas G-bound hAgo2 complexes are incredibly stable (*Olejniczak et al., 2013*). Differences between the G-bound and GT-bound states were more subtle. Significant differences were localized mainly to regions expected based on crystal structures, including the L1 linker and helix 7 of the L2 linker region (*Figure 4*), which undergo a rigid body movement upon target binding, widening the N-PAZ channel to accommodate pairing beyond the seed sequence (*Schirle et al., 2014*).

## Phosphorylation occurs hierarchically from S828

We next sought to determine where in the EI phosphorylation was occurring. CK1α has a canonical sequence preference of pS/pT-x-x-S/T* or (E,D)$_n$-x-x-S/T*-x where 'p' indicates a phosphorylated site,

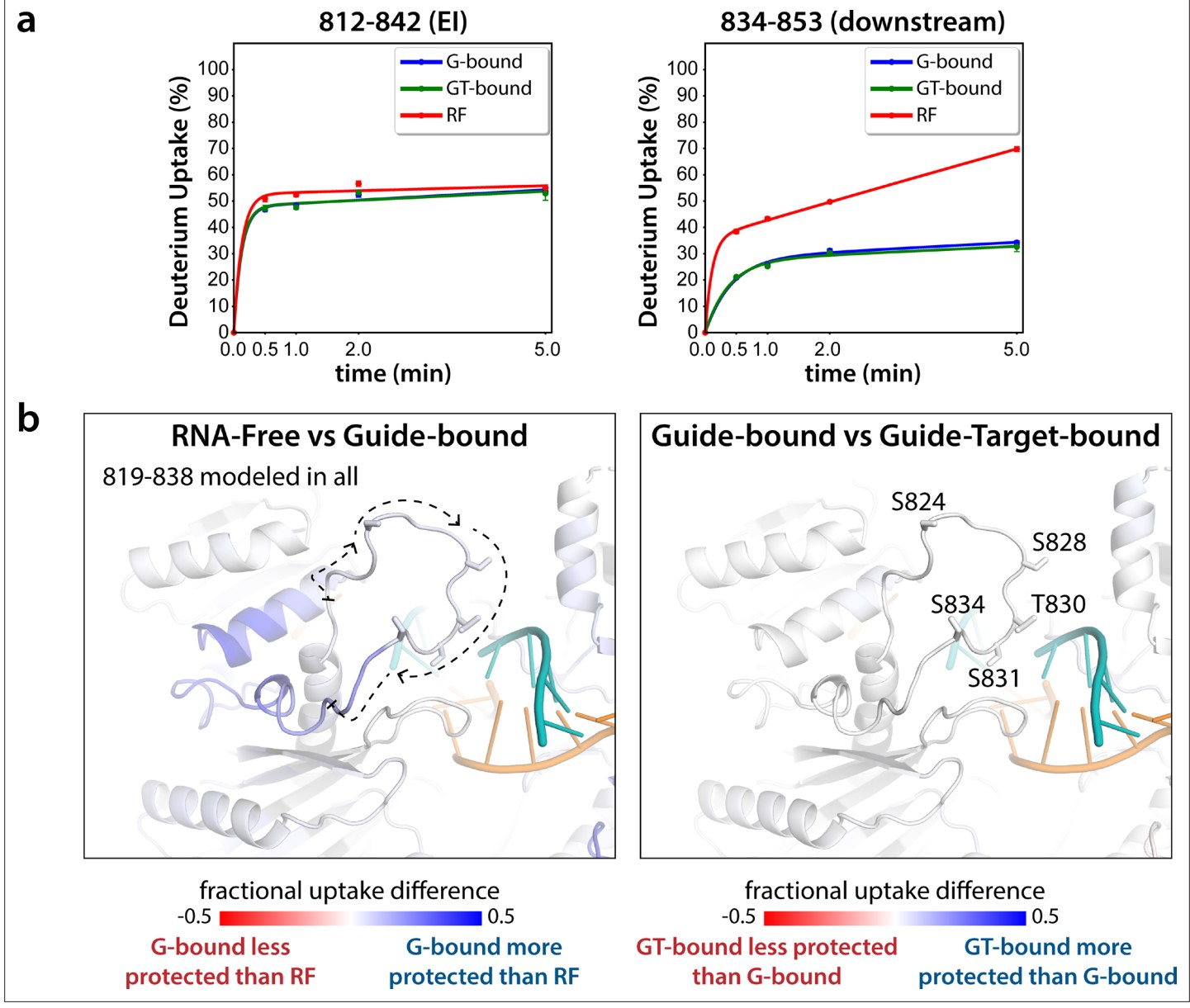

**Figure 3.** Hydrogen-deuterium exchange mass spectrometry (HDX-MS) of the hAgo2(5XA) eukaryotic insertion (EI). (**a**) Deuterium uptake plots for the shortest peptide covering the EI shows no significant difference between the guide-bound (**G-bound**) and guide + target-bound (GT-bound) states. The increased uptake in the RNA-free (RF) state can be accounted for by increased uptake in region downstream of the phosphorylation sites. (**b**) Fractional uptake differences between the RF and G-bound (left) or the G-bound and GT-bound hAgo2(5XA) (right). Residues 819–838 were modeled into the structure of hAgo2 bound to a target with seed and supplementary pairing (PDB code 6N4O) in Chimera using MODELLER's loop refine tool. Peptides were filtered for significant differences and filtered data for the 30 s timepoint are mapped onto the model structure. Phosphorylatable residues are displayed in stick representation and labeled.

The online version of this article includes the following figure supplement(s) for figure 3:

**Figure supplement 1.** 5XA substitution does not alter hAgo2's target affinity.

**Figure supplement 2.** 5XA and wild-type (wt) hAgo2 show near-identical deuterium exchange patterns, including in regions surrounding the eukaryotic insertion (EI), indicating that 5XA can serve as a proxy for the wt, for which we were unable to collect data with reliable EI coverage.

**Figure supplement 3.** Heat map showing relative deuterium uptake for wild-type (wt) hAgo2 in RNA-free (RF) state, and 5XA hAgo2 in RF, guide-bound (G-bound), and guide + target-bound (GT-bound) states across the entire protein and all timepoints.

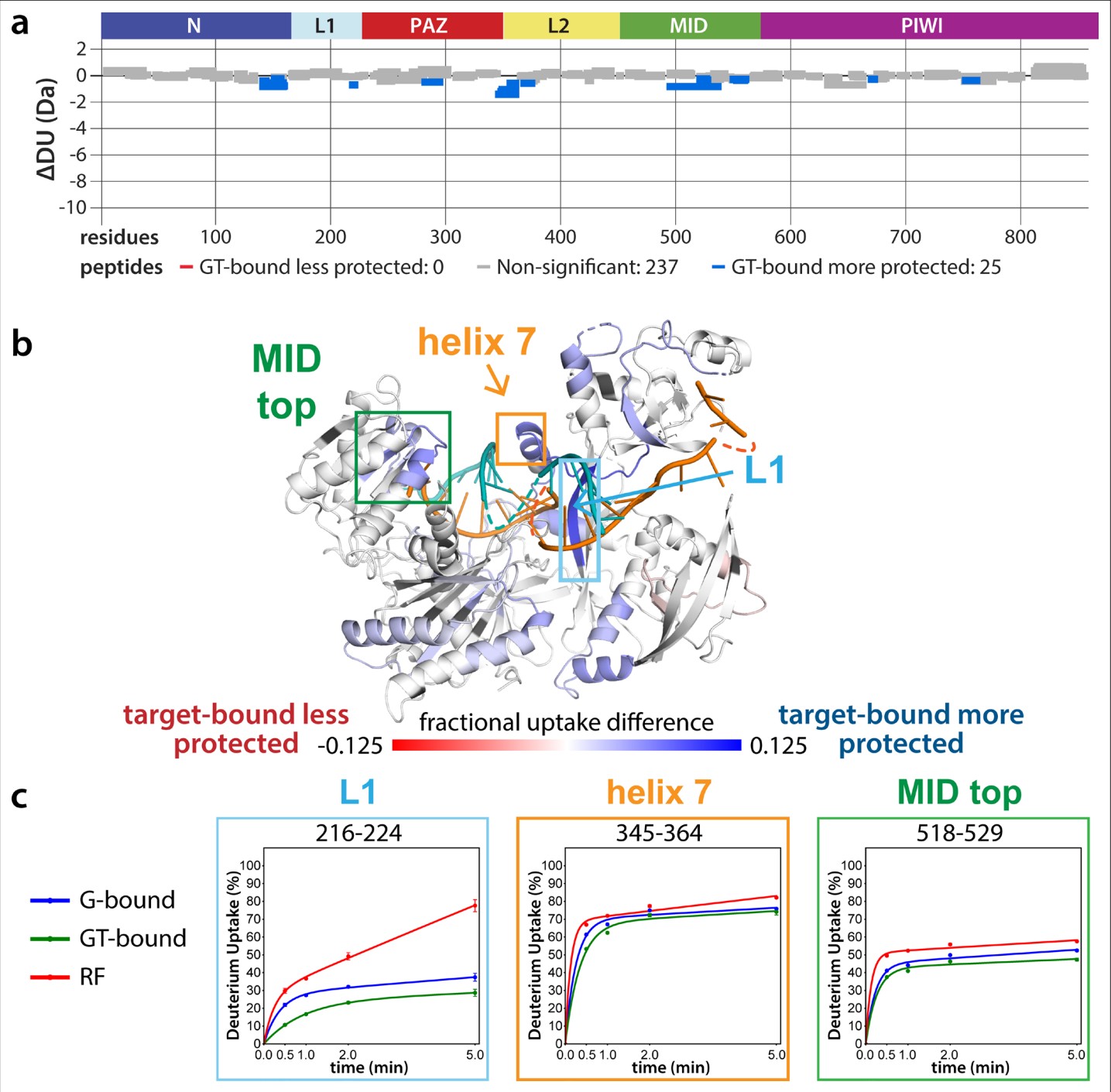

**Figure 4.** Effects of target-binding on deuterium exchange are structurally concentrated and small in magnitude. (**a**) Wood's plot comparing the absolute deuterium uptake of guide-bound 5XA hAgo2 (G-bound) and guide + target-bound 5XA hAgo2 (GT-bound) at the 30 s timepoint. Statistically different (p < 0.1) peptides are highlighted in blue indicating that the GT-bound is somewhat more protected than the G-bound. Positions of hAgo2 domains are displayed above. (**b**) Fractional uptake difference comparing G-bound and GT-bound 5XA hAgo2 at the 30 s timepoint. Peptides were filtered for significant differences and filtered data are displayed on the structure of hAgo2 bound to a guide and target with seed and supplementary pairing (PDB code 6N4O, *Sheu-Gruttadauria et al., 2019b*; *Sheu-Gruttadauria et al., 2019a*), scaled to the data range (dark red where target-bound is 12.5% less protected to dark blue where target-bound is 12.5% more protected). (**c**) Deuterium uptake plots for peptides representative of three main areas showing protection offered upon target binding.

The online version of this article includes the following figure supplement(s) for figure 4:

**Figure supplement 1.** Guide-binding offers strong protection from deuterium exchange over most of hAgo2.

* indicates the site to be phosphorylated, and 'x' indicates any amino acid (*Venerando et al., 2014*). Thus, prior phosphorylation (by CK1α or another kinase) can prime for subsequent phosphorylation, making CK1α capable of hierarchical phosphorylation. This has been proposed to occur at the EI (*Golden et al., 2017*), however heterogeneity has hindered past attempts to characterize EI phosphorylation status at the amino acid level. In addition to differences in phosphorylation status among Ago molecules, identical charged states (e.g. 1P, 2P, 3P, or 4P) can be formed with different combinations of phosphorylated residues. Furthermore, due to the close clustering of the sites, as well as the tendency of phosphorylations to be lost during processing and mass spectrometry, proteomic studies have been unable to identify every phosphorylation combination (*Golden et al., 2017*; *Quévillon Huberdeau et al., 2017*).

In order to determine which EI residues get phosphorylated in vitro, we addressed this ambiguity by performing in vitro phosphorylation assays on hAgo2 constructs in which we prevented phosphorylation at specific sites by mutating each potentially phosphorylatable serine or threonine to alanine, individually or in various combinations. Since we had previously determined that target binding is required for phosphorylation (*Figure 1*), we carried out these assays in the presence of guide (miR-200) and target (ZT1). We started by looking at which residues could get phosphorylated by CK1α in vitro without priming phosphorylation by way of creating a series of five 'single o' hAgo2 mutations in which only a single residue is 'open' for phosphorylation, with the others changed to alanine. For example, to check for S824 phosphorylation, we designed an Ago2(S828A/T830A/S831A/S834A) mutant, abbreviated to '824o'. We observed phosphorylation of the 828o mutant, but not of any of the other four (*Figure 5a*). Thus, only S828 can get phosphorylated by CK1α in vitro in the absence of prior priming phosphorylation. An upstream acidic cluster in the EI was proposed to provide a non-primed canonical recognition sequence (*Golden et al., 2017*) however, we found that phosphorylation was not impacted when we mutated those acidic residues to neutral ones (i.e. E821Q/D823N/E826Q) (*Figure 5—figure supplement 1*).

To examine the possibility that CK1α-mediated phosphorylation of S828 may serve a priming role for additional EI phosphorylations, we first tried an S828 phosphomimetic (S828E). This failed to prime for further phosphorylation (*Figure 5—figure supplement 2*), however glutamate is different in size and charge to phosphoserine and thus may not sufficiently recapitulate the effects of true phosphorylation (*Chen and Cole, 2015*). Therefore, we engineered 'double o' hAgo2 EI constructs in which 828 and only one other residue are open for phosphorylation, while the other three sites are mutated to alanine. In these constructs, S828 becomes phosphorylated and can potentially serve to prime further phosphorylation of the other available site, resulting in increased phosphorylation compared to the singly phosphorylatable 828o mutant. The 'double o' mutant in which S828 and S831 were open (i.e. Ago2(T830A, S831A, S834A)) showed significantly increased phosphorylation, whereas the mutants with 824, 830, or 834 open showed only minor increases (*Figure 5b*). This suggested that pS828 is capable of priming for phosphorylation of S831.

To determine whether pS831 is subsequently capable of priming for further phosphorylation, we created 'triple o' mutants: 824o;828o;831o [Ago2(T830A;S834A)]; 828o;831o;834o [Ago2(S824A;T830A)]; and 828o;830o;831o [Ago2(S824A;S834A)]. We saw an increase in phosphorylation in all three cases, suggesting that, after the second phosphorylation at S831, the EI phosphorylation pattern can follow different routes and is more heterogenous than the initial S828 followed by S831 phosphorylation pattern (*Figure 5c*). Individual mutation of S824, S828, or S834 to alanine did not significantly affect phosphorylation compared to the wt, suggesting that, in our in vitro system, multiple third/fourth sites can be phosphorylated but likely never all five at the same time. An alternative explanation is that only 828 and 831 get phosphorylated in our system; however, the 'triple o' mutants show a lower degree of phosphorylation than the wt (*Figure 5c*). Therefore, this alternative explanation would imply that alanine mutations at two sites make the EI a worse substrate overall, decreasing 828 and/or 831 phosphorylation. Importantly, mutation of S828 to alanine prevented almost all detectable phosphorylation and mutation of S831 to alanine substantially decreased phosphorylalation (*Figure 5d*), supporting their key roles in the hierarchy of EI phosphorylation. We therefore favor the first possibility that not all five are phosphorylated.

## Phosphorylation promotes target release

Next we wanted to study the functional consequences of this phosphorylation on the interaction of minimal RISC (hAgo2 + guide) with target RNA. Because, as shown above, phosphorylation can only be

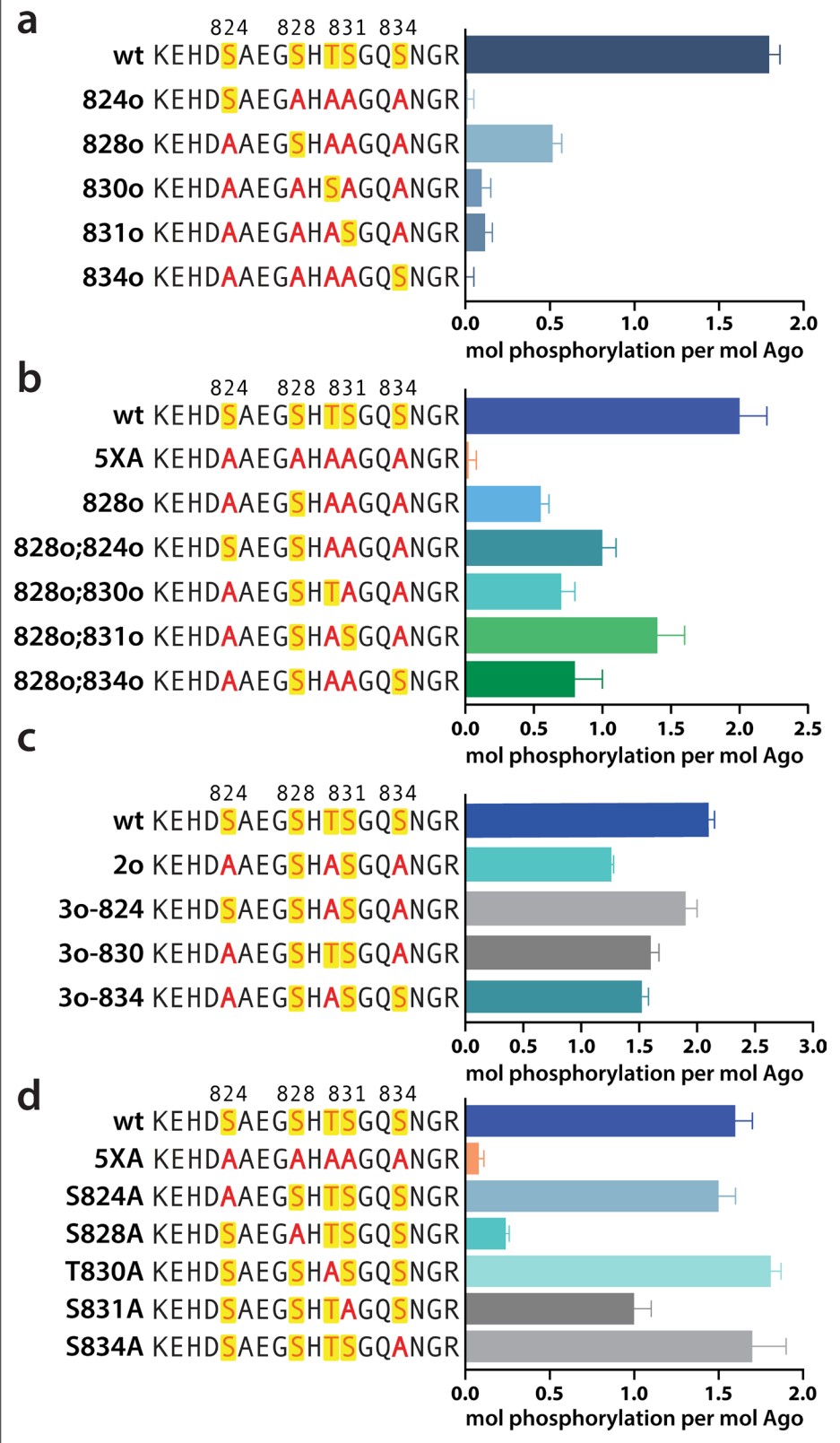

**Figure 5.** Phosphorylation of the hAgo2 eukaryotic insertion (EI) occurs hierarchically. (**a**) CK1α-mediated phosphorylation of miR-200/ZT1-bound wild-type (wt), 5XA, and 'single open' mutants in which only a single potential phosphosite is open for phosphorylation by CK1α, with the rest mutated to alanine, shows that S828 is the only site capable of CK1α-mediated phosphorylation without previous priming phosphorylation. (**b**) 'Double

*Figure 5 continued on next page*

*Figure 5 continued*

open' mutants in which only two sites are open for phosphorylation, S828 and one other site, show that S828 is capable of priming for S831 phosphorylation. (**c**) 'Triple open' mutants in which only three sites are open for phosphorylation, S828, S831, and S834, show that S831 is capable of priming for further phosphorylation, but that the third site phosphorylation is more heterogeneous. (**d**) Similar effects of single alanine substitutions for S824, T830, and S834 underscore the heterogeneous nature of the third phosphorylation site. Quantification of phosphorylation was done by liquid scintillation and comparison to a spotted ATP standard. Data are presented as the mean ± standard error (SE) for three technical replicates.

The online version of this article includes the following figure supplement(s) for figure 5:

**Figure supplement 1.** The cluster of upstream acidic residues does not play a priming role.

**Figure supplement 2.** Phosphomimetics failed to prime for phosphorylation.

achieved *after* target binding, we conducted target release assays in which we phosphorylated hAgo2 bound to the guide RNA and radiolabeled target RNA (ZT1). We then used a time course filter-binding assay to measure release after dilution into a 250-fold excess of unlabeled target. CK1α-mediated phosphorylation of wt hAgo2 significantly increased $k_{off}$ by ~7.5-fold, and thus decreased the half-life of the RISC/target complex (*Figure 6a*). These effects are specific to EI phosphorylation, as CK1α-mediated treatment of hAgo2(5XA) had no effect. These effects are additive – the more sites available for phosphorylation, the larger the observed effect (828o < 828o;831o < 828o;831o;834o) (*Figure 6b*). The 'triple o' mutant 824o;828o;831o showed similar effects as 828o;831o;834o (*Figure 6—figure supplement 1*), suggesting that the overall charge, rather than the specific phosphorylated sites, is most important for target release.

It was proposed that the negative charge of the added phosphates could clash with the negative charge of the target RNA backbone, leading to loosening of hAgo2 binding to the target by charge repulsion (*Quévillon Huberdeau et al., 2017*). To test this, we repeated the release assay with high salt concentrations, where the high ionic strength should provide electrostatic shielding, thus minimizing charge-based effects. In these conditions, differences between the off-rate of CK1α-treated and untreated wt hAgo2 were much smaller, only ~1.5-fold higher for phosphorylated hAgo2, supporting the hypothesis that the effects of phosphorylation on promoting target release are largely electrostatically mediated (*Figure 6c*).

## Discussion

miRNA-mediated gene repression is a widespread mode of regulation in metazoans. Most human genes contain at least one evolutionarily conserved miRNA target site, and many genes are under regulation by multiple families of miRNAs. Each miRNA family, in turn, typically has hundreds of targets; the 90 most broadly conserved miRNA families have over 400 conserved targets each (*Friedman et al., 2009*). At the same time, the number of miRNA target sites is much greater than the number of miRNA-Ago complexes (*Denzler et al., 2014*) and the half-lives of mRNAs are, on average, significantly shorter than those of RISC complexes (*Kingston and Bartel, 2019*). Therefore, it appears that each core RISC targets multiple mRNAs throughout its lifetime. On the other hand, repression efficacy is largely related to binding stability, with stronger binding correlating with enhanced repression (*McGeary et al., 2019*). Thus, ensuring robust mRNA repression at the global level is a balancing act between enabling strong binding at individual sites and binding to a large number of sites. 3' supplementary pairing increases binding affinity and thus can lead to enhanced repression at cognate sites (*Becker et al., 2019*), but high affinity may come with the risk of effectively sequestering RISC. CK1α-mediated EI phosphorylation provides a mechanism to counterbalance this added affinity, freeing RISC to repress additional targets. Indeed, Golden et al. found that hAgo2 which could not be phosphorylated in the EI associated with a more diverse pool of mRNA targets, but with decreased coverage of individual targets, accompanied by global dysregulation of miRNA-mediated repression (*Golden et al., 2017*).

This may help explain discrepancies between the in vitro and in vivo effects of 3' supplementary pairing. In vitro, 3' supplementary pairing can significantly enhance the stability of Ago/target interactions, with 3–4 contiguous canonical base pairs in the supplementary region increasing affinity over 20-fold for some miRNA/target pairs (*Sheu-Gruttadauria et al., 2019b*; *Xiao and MacRae, 2020*;

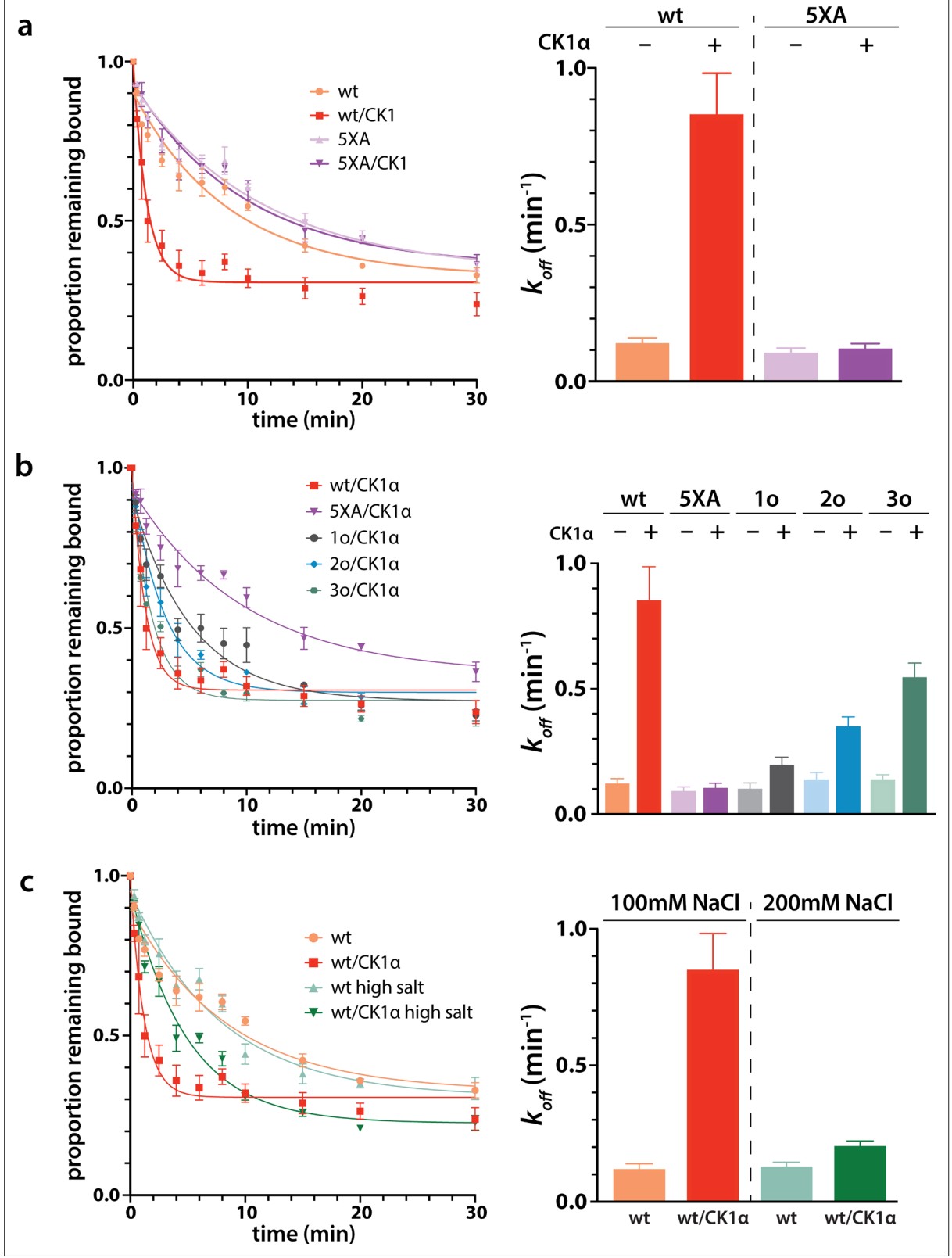

**Figure 6.** Phosphorylation promotes target release. (**a**) Filter-binding-based target release assay tracking the release of labeled target from wild-type (wt) hAgo2 phosphorylated while bound to unlabeled target. (**b**) Target release assays performed as in a with 1 open, 2 open, and 3 open (824o;828o;831o) hAgo2 mutants show that increasing phosphorylation decreases half life. (**c**) The effects of phosphorylation on target release are minimal under high salt conditions (200 mM NaCl vs. 100 mM NaCl), showing that the effect on target release is largely electrostatic. (**a–c**) Proportion

*Figure 6 continued on next page*

*Figure 6 continued*

of target remaining bound over time since addition of an unlabeled target chase is plotted with 95% CI (left) and half lives calculated by fitting to a single-phase exponential decay equation are shown ± standard error (SE) (right). n = 4 technical replicates for 3o-834/CK1α, n = 3 technical replicates for others.

The online version of this article includes the following figure supplement(s) for figure 6:

**Figure supplement 1.** Different triple open combinations show similar decreases in target half life in response to phosphorylation.

*Salomon et al., 2015*). This effect is more pronounced for miRNAs with weak, low GC-content, seeds (*Xiao and MacRae, 2020*; *Salomon et al., 2015*). However, several reports concluded that the contributions of supplementary pairing in vivo and in cell culture are less substantial (*Grimson et al., 2007*; *Wee et al., 2012*; *Salomon et al., 2015*). Some of this discrepancy has been attributed to differences in cell types and measurement conditions, as well as the large effects of a small number of miRNA/targets being diluted out in large datasets (*Xiao and MacRae, 2020*). Additionally, the precise position and composition of optimal pairing in the 3' region has been found to differ for different miRNAs (*McGeary et al., 2022*), complicating classifications and predictions.

However, as we have shown in this study, supplementary pairing triggers phosphorylation of the EI, which could offset added binding affinity. Therefore, in vivo, the overall effects of supplementary pairing on target repression would appear to be less pronounced. This would be consistent with supplementary pairing functioning primarily to distinguish between target sites for the same miRNAs. Phosphorylation of targets with supplementary pairing *after* repression is established, freeing RISC to repress additional targets, would allow for the most efficient use of supplementary pairing for such discriminatory purposes.

CK1α phosphorylation likely occurs in a distributive manner (*Gebel et al., 2020*). Because phosphorylation of multiple residues would require CK1α to actively interact with Ago multiple times, supplementary pairing might be expected to increase phosphorylation by way of added affinity. However, we found that target retention time is not the primary driver of CK1α-mediated EI phosphorylation, as demonstrated by the minimal levels of phosphorylation observed upon binding to the fully complementary or TDMD-like target. Our findings instead suggest that the conformation of the RISC complex is of primary importance with regard to triggering phosphorylation. Compared to 3' supplementary targets, fully complementary targets require different conformational changes due to base pairing of the central region (*Sheu-Gruttadauria et al., 2019b*), and these changes might make Ago's EI a poorer substrate for CK1α.

Recently, a study linked a variety of germline single amino acid mutations in hAgo2 to RNAi deficits and neurological symptoms (*Lessel et al., 2020*). These mutations clustered in helix 7, L1, and a loop in the PIWI domain, and, when expressed in cell culture, some of the mutated Agos had decreased EI phosphorylation and a slight increase in quantity of select target mRNAs. This, combined with molecular dynamics simulations, led the authors to propose that reduced phosphorylation prevents efficient target release. Based on our findings that target binding promotes phosphorylation, prolonged target binding in the usual conformation would be expected to increase rather than decrease phosphorylation. Therefore, it appears possible that the identified mutations may cause hAgo2 to adopt a suboptimal target-bound conformation which disfavors, or at least fails to promote, phosphorylation.

There are multiple examples in the literature where substrate tertiary structure has been shown to be critical for promoting CK1-mediated phosphorylation. For example, T-antigen is phosphorylated by CK1 at non-canonical sites at its N-terminus in a manner that is dependent on proper folding and an intact C-terminus (*Cegielska et al., 1994*). Similarly, conformational changes induced by target binding may cause Ago to adopt a conformation which is structurally preferable for CK1α. However, this preference cannot solely be attributed to the changes making the EI more accessible. Despite the high degree of flexibility and solvent accessibility of the EI and the surrounding area in the RNA-free hAgo2, as seen from our HDX-MS results (*Figure 3* and *Figure 4—figure supplement 1*) and through limited proteolysis (*Elkayam et al., 2012*), we were unable to phosphorylate it in vitro (*Figure 1*). This suggests there must be an additional factor required. Since the EI shows no significant changes in deuterium exchange neither upon guide- nor upon target-binding, the additional information needed to enable phosphorylation stems from outside of the EI. Such information could come from a different part of the hAgo2 protein that is now in a more suitable conformation to interact with the kinase.

Alternatively and/or additionally, factors outside of hAgo2 may assume this role, potentially the target RNA itself.

As we show in this study, target binding triggers the CK1α-mediated phosphorylation of S828 (*Figure 1*). However, this residue is a non-canonical target for CK1α, deviating from its preferred consensus sequence, as determined largely through peptide-based studies. Numerous examples have now come to light in which CK1α phosphorylates non-canonical sites, and additional consensus sequences have been identified (*Marin et al., 2003*). Nevertheless, CK1 kinases are thought to generally prefer primed substrates. Binding of the phosphorylated, priming residue, or a cluster of acidic residues to a basic patch on the kinase, named Anion Binding Site 1, is thought to stabilize the kinase in an active conformation and position the downstream phosphorylation site in the active site (*Gebel et al., 2020*; *Philpott et al., 2020*; *Longenecker et al., 1996*). However, phosphorylation was not impacted when we neutralized the acidic residues previously proposed to serve as a priming cluster (*Golden et al., 2017*; *Figure 5—figure supplement 1*). We would like to suggest that it is the target RNA's backbone, being negatively charged, that may be able to bind to this site and prime for S828 phosphorylation. The need for seed + supplementary pairing as opposed to other complementarity combinations in order to trigger robust phosphorylation supports a role for conformational change. These changes might position the RNA to interact with the kinase properly, rather than making the EI more accessible.

The need for target binding could explain findings from previously reported in vitro phosphorylation assays testing CK1α's ability to phosphorylate peptide substrates spanning hAgo2's EI region (*Golden et al., 2017*). Golden et al. reported that an unphosphorylated EI peptide could not be phosphorylated by recombinant CK1α, leading them to propose that contextual features in the full-length hAgo2 were required in order to make S828 a suitable CK1α substrate. Our work shows that it is not only full-length hAgo2 that is required, but rather hAgo2 bound to a guide and target RNA with seed and supplementary region pairing.

All the in vitro phosphorylation experiments that have been reported point to a central importance of S828. Our finding that S828 phosphorylation is required for subsequent hierarchical phosphorylation by CK1α in full-length hAgo2 (*Figure 5*) is consistent with previous experiments using phosphorylated peptide substrates (*Golden et al., 2017*). Golden et al. showed that peptides phosphorylated at S828 were able to be phosphorylated at S831 and those phosphorylated at S831 were able to be phosphorylated at S834. They suggested that pS828 primes for hierarchical phosphorylation of the EI at S831 and S834, consistent with our experiments with full-length hAgo2. This was further supported by their finding that S828A mutation, but not a single A mutation at any of the other phosphorylatable residues in the EI, eliminated detectable phosphorylation of hAgo2 in HCT116 cells lacking expression of ANKRD52, a regulatory subunit of PP6 (*Golden et al., 2017*). The central importance of S828 is also reflected in functional data from in vivo experiments. Huberdeau et al. found that mutating the corresponding serine in the *C. elegans* hAgo2 homolog ALG-1 (S992) to alanine led to severe developmental defects similar to those seen when mutating all EI phosphorylation sites to alanine (*Quévillon Huberdeau et al., 2017*). Furthermore, Golden et al. reported that expression of hAgo2(S828A), but not wt hAgo2, was able to rescue repression of a miR-19 EGFP reporter in ANKRD52-deficient cells. Phosphorylation of S828 thus appears to serve as a key gatekeeper regulating further phosphorylation and its consequent functional effects, implying its regulation would be critical. We now show that this gatekeeper is directly regulated by target binding.

Because miRNA-mediated repression does not involve slicing of the targets, additional cofactors are required to carry out silencing of miRNA targets. After miRNA-bound Ago binds a target mRNA, Ago binds a scaffolding protein of the GW182 family (TNRC6A/B/C in humans), which in turn recruits deadenylation complexes (PAN2-PAN3 and/or CCR4-NOT) and decapping complexes to promote target degradation (*Jonas and Izaurralde, 2015*). Previous studies have found that EI phosphorylation does not impact hAgo2/GW182 interaction (*Golden et al., 2017*; *Quévillon Huberdeau et al., 2017*). It is possible that one of the recruited proteins recruits CK1α. However, since we are able to observe Ago phosphorylation in the absence of additional proteins, they are not required for CKα-mediated phosphorylation. Importantly, because we did not observe phosphorylation until after target binding, and even then, can only detect initial phosphorylation of a single site, the in vitro system appears to recapitulate a regulated system having high specificity.

Because Ago must be dephosphorylated by PP6 before it can optimally repress additional targets, phosphorylation also has the potential to serve as a global regulatory measure. Knocking out PP6 in human cell lines led to a global decrease in RISC binding to mRNA (*Golden et al., 2017*). Under normal conditions, however, CK1α-mediated EI phosphorylation is rapidly countered by dephosphorylation (*Golden et al., 2017*). Despite this, neither 5XA nor 5XE hAgo2 could rescue miRNA-mediated repression in Ago2$^{-/-}$ cells, indicating that the phosphorylation cycling process itself is important, not merely the presence or absence of phosphorylation.

Our findings that *only* S828 can be phosphorylated in our in vitro system without priming phosphorylation, and that this phosphorylation only occurs after binding to targets meeting certain criteria reveal novel structural and sequence determinants of highly specific CK1α-mediated phosphorylation. We therefore propose that target binding serves as a means for regulating CK1α-mediated phosphorylation of Ago, which, in turn, regulates RISC turnover and thus miRNA-mediated mRNA repression. The presence of multiple phosphorylation sites with decreasing target affinity may merely reflect the biophysical requirements for establishing sufficient negative charge to repel the target RNA. However, it might also provide a way for a cell to titrate the Ago/target affinity and/or to serve as a 'checkpoint' in order to ensure that targets aren't spuriously released. Our work doesn't address the temporal dynamics of phosphorylation, dephosphorylation, and repression. However, given the previous in vivo findings showing widespread regulatory defects with both hyper- and hypophosphorylation (*Golden et al., 2017*; *Quévillon Huberdeau et al., 2017*), we speculate that phosphorylation, or at least phosphorylation of enough sites to efficiently promote target release, happens after target repression is established. The requirement for multiple phosphorylations could help ensure that there is enough time for the initiation of repression to occur before the target is released. Direct tethering of GW182 family proteins to mRNA can bypass the need for RISC (*Behm-Ansmant et al., 2006*; *Zipprich et al., 2009*), suggesting that if enough Ago-independent interactions between repressive machinery and the mRNA are formed, Ago may even be able to be released while repression is ongoing.

This mechanism appears specialized for repression mediated by miRNA, as opposed to siRNA, as binding to fully complementary sites only triggers minimal phosphorylation (*Figure 2b*) and phosphomimicking does not impair slicing (*Figure 2—figure supplement 2*). This is consistent with the evolutionary conservation of the phosphosites among miRNA-specialized Ago proteins (e.g. *D. melanogaster* Ago1 [dmAgo1]) but not among siRNA-specialized Agos (e.g. dmAgo2), and our work provides a mechanistic explanation. With siRNA, slicing of the fully complementary target promotes its release by destabilizing the guide/target duplex (*Salomon et al., 2015*), but partially-complementary miRNAs with high affinity, such as those with 3' supplementary pairing, could get stuck on a target and phosphorylation could serve as a mechanism to promote their release in the absence of slicing. However, we found that extending 3' pairing in the absence of central pairing also decreased EI phosphorylation (*Figure 2b*). Such targets can induce TDMD in vivo, which resolves the interaction through degradation of the RISC complex (*Shi et al., 2020*; *Han et al., 2020*), and therefore these targets would not need phosphorylation-promoted release to resolve RISC/target interactions.

Though we showed that hAgo1 and hAgo3 are also subject to target-binding-triggered CK1α-mediated phosphorylation in vitro similar to hAgo2, we did not investigate the functional impact of this phosphorylation; however, based on the high homology between the human Ago's and their apparent redundancy in most instances (*Su et al., 2009*), we would expect the effects of hAgo1 and hAgo3 phosphorylation to be similar to those we observed with hAgo2. Therefore, this regulatory mechanism is likely shared by these miRNA-handling Ago orthologs, suggesting a potentially broad impact on post-transcriptional regulation.

In conclusion, our findings show that Ago EI phosphorylation is a post-target-binding regulatory mechanism that promotes target release, freeing guide-bound Ago, namely RISC, from the mRNA target and dissuading it from binding more targets until it gets dephosphorylated by PPP6C/ANKRD52 (*Golden et al., 2017*). This dephosphorylation was previously shown to be rapid (*Golden et al., 2017*), therefore the released Ago-guide complex is likely to quickly be able to seek out and repress additional targets. Although we did not investigate in vivo temporal dynamics, we suggest that phosphorylation occurs once repression has already been established and Ago's presence is no longer required. As long as the cycle of phosphorylation/dephosphorylation is not disrupted, this would allow cells to strike a balance between the increased efficacy and specificity provided by stronger binding sites and the quest for efficiency, allowing a single miRISC to repress multiple mRNAs (*Figure 7*). This

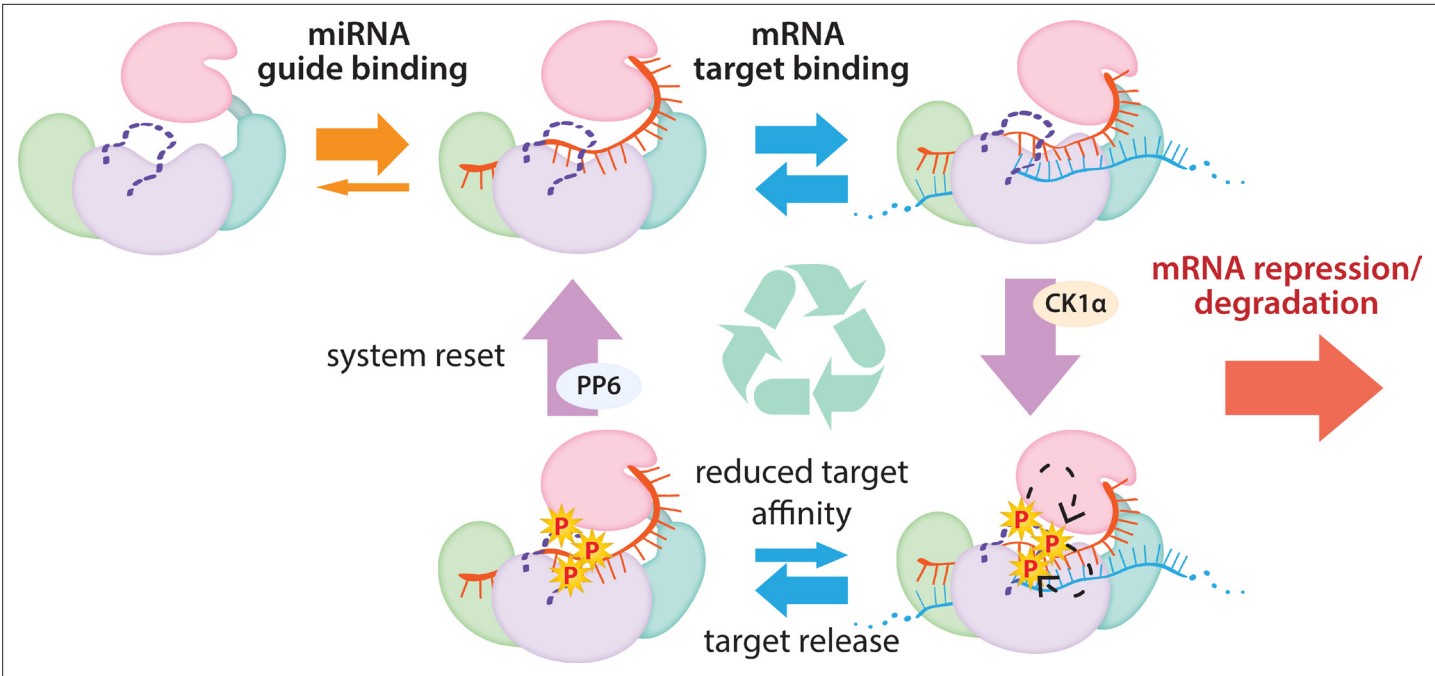

**Figure 7.** The cycle of phosphorylation/dephosphorylation plays a regulatory role in silencing. In order to repress a target mRNA, Ago binds to a small RNA guide to form the core RISC complex, which subsequently binds to target mRNAs containing sequence complementarity to the guide to effect repression. The core RISC complex (Ago/guide) is incredibly stable but would have to release target mRNA prior to repression of additional targets. We propose that this release is facilitated by CK1α-mediated phosphorylation of the eukaryotic insertion (EI). We showed that phosphorylation is triggered by binding to targets that are not fully complementary, but that contain complementarity to the seed and 3' supplementary pairing regions of the guide. Binding to such targets results in the complex adopting a productive conformation for EI phosphorylation. We suggest this may be due to orienting of the RNA backbone phosphates to serve as non-canonical priming phosphates. EI phosphorylation occurs hierarchically starting at S828 and the additive negative charges of the phosphates electrostatically repel the target RNA, leading to target release. The phosphorylated RISC has reduced affinity for further target binding. However, the system is quickly reset by dephosphorylation. This allows the RISC complex to bind to and repress additional sites, helping to explain how RISC can function efficiently and effectively in the context of excess target sites.

mechanism of promoting miRNA target release appears specialized for miRNA-mediated repression and the broad conservation of the phosphorylation sites suggests that its use is widespread among miRNA-handling Ago proteins.

## Materials and methods
### Protein expression and purification
Recombinant hAgo2 was prepared as previously described (*Elkayam et al., 2012*), with the addition of a dephosphorylation step. Briefly, hAgo2 was expressed in Sf9 insect cells (Life Technologies/Thermofisher) with an N-terminal StrepII_SUMO fusion tag using the MultiBac system (*Fitzgerald et al., 2006*). Sf9 cells were routinely checked for mycoplasma contamination. After Strep-Tactin affinity chromatography (Strep-Tactin Superflow High Capacity, IBA) (50 mM Tris pH 8, 100 mM KCl, 5 mM DTT), the tag was removed using TEV protease and RNA-free Ago was separated from Ago loaded with endogenous (insect) RNAs by cation exchange chromatography with a stepwise KCl gradient elution (MonoS 10/100, Cytiva). RNA-free Ago was treated overnight at 4°C with $\lambda$ protein phosphatase ($\lambda$ PP) at a 1:2.5 molar ratio, with $MnCl_2$ added to a final concentration of 1 mM. The protein was then further purified via size exclusion chromatography (Superdex 200 10/300 Increase GL, Cytiva) (10 mM Tris pH 8, 200 mM KCl, 5 mM DTT). Aliquots were frozen at –80°C, 2 mg/mL, 10% glycerol. hAgo1 and hAgo3 were prepared in a similar manner.

Human CK1α (isoform 1) was expressed in Sf9 insect cells with an N-terminal StrepII_SUMO fusion tag using the MultiBac system. Following Strep-Tactin affinity chromatography (50 mM Tris pH 8, 200 mM NaCl, 5 mM DTT), the tag was removed using TEV protease and CK1α was further purified by cation exchange chromatography (MonoS, linear NaCl gradient elution) and size exclusion

chromatography (Superdex75 Increase 10/300 GL, Cytiva) (10 mM Tris pH 8, 200 mM NaCl, 5 mM DTT). Aliquots were frozen at –80°C, 0.5 mg/mL, 50% glycerol.

$\lambda$ PP was cloned into a pET28 vector with an N-terminal His tag and expressed in BL21(DE3) *E. coli*. It was induced with 1 mM IPTG and expressed overnight at 18°C in TB media. It was then purified by Ni-NTA affinity chromatography (Qiagen) (50 mM Tris pH 8, 300 mM NaCl, 25 mM imidazole, 0.5 mM TCEP, 0.1 mM $MnCl_2$; eluted with 250 mM imidazole) followed by size exclusion chromatography (Superdex75) (50 mM Tris pH 8, 150 mM NaCl, 0.5 mM TCEP, 0.1 mM $MnCl_2$). Activity was verified using a colorimetric pNPP assay and aliquots were frozen at –80°C in 50% glycerol, 2.7 mg/mL.

## In vitro phosphorylation assays

In vitro phosphorylation was carried out with hAgo2:guide:target at 1 µM and CK1α at 20 nM. Prior to adding CK1, Ago was incubated 1:1 with guide RNA for 30 min, followed by incubation with target RNA (1:1:1 molar ratio) for 30 min, both steps were carried at room temperature (RT). The complex was then mixed with kinase reaction buffer (final concentrations of 200 µM total ATP, 2 nM [γ-32P]-ATP (Perkin-Elmer), 25 mM Tris pH 7.4, 10 mM $MgCl_2$, 2.5 mM DTT, 0.5 mM $Na_3VO_4$). CK1α was then added and reactions were incubated at 37°C for 90 min at which time they were quenched with a stop buffer containing 25 mM EDTA and 25 mM unlabeled ATP. Aliquots were spotted on phosphocellulose paper, washed three times (5 min each) in 75 mM phosphoric acid, one time in acetone, and then air-dried and measured using liquid scintillation counting. Moles of phosphate per aliquot were calculated by comparison to spotted (unwashed) reaction buffer. This value was divided by the number of moles of Ago in the aliquot to determine the mole phosphorylation detected per mole Ago.

## Target binding

Equilibrium target-binding assays were performed using a double-membrane filter-binding setup with a slot blot vacuum filtration device (Bio-Rad SF) containing an upper nitrocellulose membrane (to capture protein and protein/RNA complexes) and a lower nylon membrane (to capture free RNA) as previously described. Briefly, hAgo2 and guide were mixed at a 1:1 molar ratio and incubated at RT for 30 min, then serially diluted in 10 mM Tris pH 8, 100 mM NaCl, 5 mM DTT, and added to 5' $^{32}$P-labeled target RNA (50–100 pM). After 1 hr incubation to come to equilibrium, protein (including RNA-bound protein) and free RNA were separated by slot blot. Results were obtained through phosphorimaging (Typhoon FLA 7000, Cytiva) and quantified using GeneTools (SynGene). Kinetic parameters were fit using GraphPad Prism (Version 9), one site-specific binding.

For competition target-binding assays, hAgo2 was mixed 1:1 with miR-200 and allowed to bind for 30 min at RT. This RISC complex was then added to a serial dilution mixture of radiolabeled ZT1 (30 nt), which was held constant at 50 pM, and unlabeled competitor target which was 1:3 serially diluted, starting at 750 nM for the seed-only complementary target (ZT1.som) and 500 nM for all others. The mixture was allowed to equilibrate for 1 hr at RT before separating protein from free RNA using a filter-binding assay as above. Data was quantified in GeneTools (SynGene) and Ki was fit using GraphPad Prism.

## Guide-binding

hAgo2 and guide were mixed at a 1:1 molar ratio and incubated at RT for 60 min, then serially diluted in 10 mM Tris pH 8, 100 mM NaCl, 5 mM DTT, and added to 5' $^{32}$P-labeled miR-200 (25 pM). After 1 hr incubation to reach equilibrium, protein and protein-RNA complex were separated from free RNA by slot blot and quantified as above.

## Target release

hAgo2 was mixed 1:1 with miR-200 guide RNA and incubated for 30 min at RT. A mix of unlabeled and $^{32}$P-labeled ZT1 target RNA (30 nt) was added and incubated for a further 30 min. In vitro phosphorylation was then carried out with 500 nM hAgo2/guide, 200 nM labeled ZT1, 300 nM unlabeled ZT1, and 20 nM CK1α for 60 min at 37°C followed by 30 min at 10°C. An aliquot was then diluted with a high concentration of unlabeled target to a final concentration of 5 nM hAgo2, 2 nM labeled ZT1, 500 nM unlabeled ZT1, in either standard buffer (10 mM Tris pH 8.0, 100 mM NaCl, 5 mM DTT) or high salt buffer (10 mM Tris pH 8.0, 200 mM NaCl, 5 mM DTT) at 10°C. An initial sample was taken immediately after mixing (at 15 s) and set as the '0' timepoint. Aliquots were taken out to 30 min and

proportions bound were compared to the proportion bound in the '0' sample. At each timepoint, free and protein-bound RNA were separated via slot blot, followed by autoradiography and quantification using GeneTools. Data were fit to a One-Phase Exponential Decay equation using GraphPad Prism and plotted ± SE. Reactions carried out in triplicate.

## Slicer assays

Slicer assays were carried out as previously described with slight modifications (*Faehnle et al., 2013*). Briefly, reactions were performed in 10 mM Tris-HCl (pH 8.0), 100 mM KCl, 10 mM DTT, 2 mM MgCl$_2$, and 10% glycerol. hAgo2 protein was loaded with at miR-200 at a 1:1 ratio and incubated for 30 min at at RT. 20 µL slicing reactions were initiated by mixing 5' $^{32}$P-labeled target RNA (2 nM final concentration) in the reaction buffer with 8 nM loaded hAgo2:miR200 complex. After 60 min, the reaction quenched in formamide loading buffer and incubated at 95°C for 5 min. The radiolabeled slicing products were separated using urea PAGE, visualized by phosphorimaging, and quantified with GeneTools.

## HDX-MS

HDX-MS was carried out by the HDX Core of the University of California, San Diego Biomolecular/Proteomics Mass Spectrometry Facility (UCSD BPMSF) (NIH shared instrumentation grant S10 OD016234). Hydrogen/deuterium exchange mass spectrometry (HDX-MS) was performed using a Waters Synapt G2Si equipped with nanoACQUITY UPLC system with H/DX technology and a LEAP autosampler. Individual proteins were purified by size exclusion chromatography in 10 mM Tris pH 8, 200 mM KCl, 5 mM DTT, 10% glycerol immediately prior to analysis. The final concentrations of proteins in each sample were 5 µM. For each deuteration time, 4 µL complex was equilibrated to 25°C for 5 min and then mixed with 56 µL D$_2$O buffer (10 mM Tris pH 8, 200 mM KCl, 5 mM DTT, 10% glycerol in D$_2$O) for 0, 0.5, 1, 2, or 5 min. The exchange was quenched with an equal volume of quench solution (3 M guanidine hydrochloride, pH 2.66).

The quenched sample (50 µL) was injected into the sample loop, followed by digestion on an in-line pepsin column (immobilized pepsin, Pierce, Inc) at 15°C. The resulting peptides were captured on a BEH C18 Vanguard pre-column, separated by analytical chromatography (Acquity UPLC BEH C18, 1.7 µM, 1.0 × 50 mm, Waters Corporation) using a 7–85% acetonitrile gradient in 0.1% formic acid over 7.5 min, and electrosprayed into the Waters SYNAPT G2Si quadrupole time-of-flight mass spectrometer. The mass spectrometer was set to collect data in the Mobility, ESI + mode; mass acquisition range of 200–2000 (m/z); scan time 0.4 s. Continuous lock mass correction was accomplished with infusion of leuenkephalin (m/z = 556.277) every 30 s (mass accuracy of 1 ppm for calibration standard). For peptide identification, the mass spectrometer was set to collect data in MS$^E$, ESI + mode instead.

The peptides were identified from triplicate MS$^E$ analyses of 10 µM solution of protein in (10 mM Tris pH 8, 200 mM KCl, 5 mM DTT, 10% glycerol), and data were analyzed using PLGS 3.0 (Waters Corporation). Peptide masses were identified using a minimum number of 250 ion counts for low energy peptides and 50 ion counts for their fragment ions. The peptides identified in PLGS were then analyzed in DynamX 3.0 (Waters Corporation) and, for displaying uptake, the deuterium uptake was corrected for back-exchange as previously described (*Lumpkin and Komives, 2019*) using DECA Software,ver1.12 [https://github.com/komiveslab/DECA, copy archived at swh:1:rev:ec7a72e54aa1d-025342fcdcb3f61840e08cd456f, (*komiveslab, 2020*)]. The relative deuterium uptake for each peptide was calculated by comparing the centroids of the mass envelopes of the deuterated samples vs. the undeuterated controls following previously published methods (*Wales et al., 2008*). The experiments were performed in triplicate (*Supplementary file 1*).

Wood's differential plots were created in Deuteros 2.0 (*Lau et al., 2021*) using the peptide significance test setting, with peptides prior to back-exchange correction. For creating PyMol heat maps showing regions with significantly different deuterium exchange, peptides were filtered for significant differences in dynamic uptake curves using a custom R script, provided by Dr Julian Langer, which employs a two-stage t-test based on *Houde et al., 2011*, and described in *Eisinger et al., 2017*. Briefly, in the first stage, the mean deuterium uptake ± SEM for each peptide was subjected to a t-test (n = 3, p ≤ 0.05, two-sided, unpaired) and peptides that passed for at least three timepoints were taken to the second stage. In this stage, the summed differences for the peptides were subjected to a second t-test (n = 4, p ≤ 0.01, two-sided, unpaired) to test for overall uptake

differences in the dynamic uptake curve. Peptides that passed both filters were used to generate a heat map in DECA (*Lumpkin and Komives, 2019*) that was mapped onto structural models of hAgo2 in PyMol (Schrödinger, LLC).

## RNAs used (synthesized by Dharmacon)

miR-200 (hsa-miR-200b-3p):

> 5'-pUAAUACUGCCUGGUAAUGAUGA-3' validation of miR-200 ZT1 sites: https://www.nature.com/articles/ncb1722
> ZT1 (miR-200-binding site corresponds to position 1243–1250 of ZEB1 3'UTR):
> 5'-ACAUUAGCUGAUUUUUACCUAUCAGUAUUA-3'
> ZT1e3c (extended 3' complementarity): 5'-ACAUUAGCUCAUCAUUACCUAUCAGUAUUA-3'
> ZT1s (sliceable (fully complementary)): 5'-ACAUUAGCUCAUCAUUACCAGGCAGUAUUA-3'
> ZT1som (seed-only match): 5'-AAAAAAAAAAAAAAAAAAAAAACAGUAUUA-3'
> ZT1.20: 5'-AUUUUUACCUAUCAGUAUUA-3'
> ZT1.16: 5'-UUACCUAUCAGUAUUA-3'
> ZT1.14: 5'-ACCUAUCAGUAUUA-3'
> ZT1.13: 5'-CCUAUCAGUAUUA-3'
> ZT1.12: 5'-CUAUCAGUAUUA-3'
> ZT1.11: 5'-UAUCAGUAUUA-3'
> ZT1.10: 5'-AUCAGUAUUA-3'

## Acknowledgements

We thank all members of the Joshua-Tor lab for helpful suggestions and comments. We also thank Nicholas Tonks (Cold Spring Harbor Laboratory) for valuable discussions and use of equipment. Additional thanks to Sheena D'Arcy and Naifu Zhang (University of Texas at Dallas) for helpful discussions. We are grateful to Julian Langer and Jonathan Zoller (Max Plank Institute of Biophysics) for sharing an R script for statistically filtering peptides and demonstrating how to use it. We thank the CSHL Mass Spectrometry Shared Resource, which is supported by Cancer Center Support Grant 5P30CA045508. BB was supported by the NSF Graduate Research Fellowship Program and CSHL School of Biological Sciences. LJ is an investigator of the Howard Hughes Medical Institute. The HDX-MS was performed on the Synapt G2Si HDX mass spectrometer obtained by shared instrumentation grant S10 OD016234.

## Additional information

### Competing interests

Elad Elkayam: is affiliated with Ventus Therapeutics. The author has no financial interests to declare. The other authors declare that no competing interests exist.

### Funding

| Funder | Grant reference number | Author |
|---|---|---|
| Howard Hughes Medical Institute | | Leemor Joshua-Tor |
| NSF Graduate Research Fellowship Program | 1938105 | Brianna Bibel |
| NCI CSHL Cancer Center Support Grant | 5P30CA045508 | Leemor Joshua-Tor |
| NIH Shared Instrumentation Grant | S10 OD016234 | Elizabeth A Komives |

The funders had no role in study design, data collection and interpretation, or the decision to submit the work for publication.

## Author contributions
Brianna Bibel, Conceptualization, Data curation, Formal analysis, Methodology, Validation, Visualization, Writing – original draft, Writing – review and editing; Elad Elkayam, Conceptualization, Formal analysis, Investigation, Methodology, Supervision, Writing – review and editing; Steve Silletti, Data curation, Formal analysis, Resources, Software; Elizabeth A Komives, Formal analysis, Methodology, Resources, Software, Supervision, Writing – review and editing; Leemor Joshua-Tor, Conceptualization, Formal analysis, Funding acquisition, Methodology, Supervision, Writing – original draft, Writing – review and editing

## Author ORCIDs
Brianna Bibel http://orcid.org/0000-0001-8543-151X
Elad Elkayam http://orcid.org/0000-0001-9266-7002
Leemor Joshua-Tor http://orcid.org/0000-0001-8185-8049

## Decision letter and Author response
Decision letter https://doi.org/10.7554/eLife.76908.sa1
Author response https://doi.org/10.7554/eLife.76908.sa2

# Additional files

## Supplementary files
- Supplementary file 1. HDX summary data.
- Transparent reporting form

## Data availability
All raw files and final state data for the HDX-MS experiments have been deposited in the MASSive public repository at massive.ucsd.edu with the dataset number MSV000088648. Individual uptake plots can be viewed by opening the state data files with DECA which is available at https://github.com/komiveslab/DECA, (copy archived at swh:1:rev:ec7a72e54aa1d025342fcdcb3f61840e08cd456f).

The following dataset was generated:

| Author(s) | Year | Dataset title | Dataset URL | Database and Identifier |
|---|---|---|---|---|
| Bibel B, Elkayam E, Silletti S, Komives EA, Joshua-Tor L | 2022 | HDXMS of Target binding to human Argonaute-2 | https://doi.org/10.25345/C5MG4F | MassIVE, 10.25345/C5MG4F |

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
