## [Editor Report]

This paper provides well documented and solid biochemical data to show how phosphorylation of hAGO2 modulates RISC-target mRNA binding dynamics. The results explain how hAGO2 is released from a target allowing a limited pool of Ago proteins to target a very large repertoire of mRNA molecules. The new data and discussion support the key claims of the manuscript.

---

## [Decision Letter]

**Decision letter after peer review:**

Thank you for submitting your article "Target binding triggers hierarchical phosphorylation of human Argonaute-2 to promote target release" for consideration by *eLife*. Your article has been reviewed by 3 peer reviewers, including Pablo A Manavella as Reviewing Editor and Reviewer #1, and the evaluation has been overseen by Kevin Struhl as the Senior Editor. The following individuals involved in review of your submission have agreed to reveal their identity: Andrea Ventura (Reviewer #2); Jeremy Dufourt (Reviewer #3).

Essential revisions:

As you will see from the reviewer's comments we are all positive about your manuscript and the suggestions are, from my point of view, helpful and simple. I do believe you will be able to address all of them without problems. Still, if you consider that some of them are not feasible in a reasonable amount of time or that are unlikely to yield a meaningful outcome don't hesitate to point it.

You will also find some requests for in vivo experiments. We discussed this point with the reviewers and even when we agreed that these experiments would be a great addition to the paper we are conscious that it may demand way too much time and effort. If you feel that you can contribute with some experiments in this direction we encourage you to add them but those experiments are not essential for the paper acceptance.

*Reviewer #1 (Recommendations for the authors):*

The results in figure 2 are quite interesting. Overall, you suggest that EI phosphorylation favour charge repulsion to release the miRNA/target association; however, this is especially true for some pairing degree with neither an extensive pairing nor a low pairing being the ideal. Can you discuss this better? Why do you think miRNA/target resolution depend on EI phosphorylation in some case more than others? Wouldn't all the cases need a negative charge to release the interaction? How do you think extensive pairing is resolved?

Figure 5, what happens if you only mutate S828 to Alanine, leaving the rest open? Can you prevent EI phosphorylation in vitro, and ideally in vivo, by mutating only S828? Even when it may be a little bit hard to test, it would be interesting to see whether S828 priming occurs in vivo.

Would it be possible to reanalyze data from ref 41 and see whether there is a bias toward the extent of miRNA/target pairing? Based on your hypothesis and discussion, one could expect more targets with 3' supplemental pairing in non-phosphorylated AGO2 mutants than in WT? If the reasoning of your discussion paragraph starting at Line 389 is correct, the in vivo results in this mutant should mimic better the in vitro data showing an increased affinity for more extensive pairing.

The main open question here is the in vivo dynamics. When does the EI phosphorylation initiate and when the miRNA/target association is resolved? one would expect this is not immediate as the main RISC function is to suppress gene translation. Thus, quick phosphorylation and resolution wouldn't be ideal. Studying this aspect is beyond the scope of this paper and will need extensive analysis, but it would be nice if you discuss this point further.

Out of personal curiosity, do you know how is EI Serine conservation in plant' AGOs (AGO1)? miRNA/target pairing is extensive in plants, so I would expect that this mechanism is not conserved? There is no need to discuss this on the manuscript.

*Reviewer #2 (Recommendations for the authors):*

I have a few suggestions for the authors:

a) The use of purified components in an in vitro system is certainly a strength of this work, but it is also a limitation, since in vivo additional factors are known to interact with Argonaute (GW182 proteins as well as other effectors) and these interactions could significantly affect Ago phosphorylation. It would be perhaps useful to confirm the hyerarchical phosphorylation of Ago protein in a cell based experiment, for example by measuring phosphorylation of ectopically expressed Ago2 harboring various combinations of mutations in the 5 phosphorylation sites. The model predicts that the S828A and, to lesser extent, the S831A mutant should be resistant to phosphorylation, while mutants in the other three sites should still be at least partially phosphorylated.

b) In the introduction (line 136), the authors mention ChIP-seq as the method used By Golden and colleagues to show reduced coverage of individual targets in the absence of phosphorylation. I think in the Golden paper the method used was eCLIP, not ChIP-seq.

c) While this manuscript conclusively shows that S828 is the first residue to be phosphorylated by CK1 in a hierarchical fashion using full length Ago, a key role for S828 and S831 had been already suggested by Golden and colleagues using in vitro phosphorylation of various peptides corresponding to the EI region of Ago2 (Figure 4g). In fact, Golden et al., wrote: "Unphosphorylated peptide was a poor substrate for CSNK1A1 under these conditions, suggesting that initial phosphorylation of this region is facilitated by contextual features present in full-length AGO2. pS824 only weakly stimulated further phosphorylation. Prior phosphorylation of S828, however, robustly promoted phosphorylation of S831 (but not T830), while pS831 efficiently primed phosphorylation of S834. Taken together with our earlier data demonstrating a critical role for S828 in phosphorylation of AGO2 in cells (Figure 2e), these findings support a model whereby initial phosphorylation of S828, and potentially S824, stimulates efficient hierarchical phosphorylation of S831 followed by S834, rendering AGO2 incompetent for target binding until returned to an active state by ANKRD52-PPP6C phosphatase activity." It would be important to mention this in the discussion.

*Reviewer #3 (Recommendations for the authors):*

Line 165 Please explain why you choose miR-200b

Figure 1c why mutating all residue at the same time and not testing simple mutation and combination? this experiment is done in figure 5 and could maybe be positioned in figure 1.

Is hAgo2 plus guide phosphorylation in the expected background or it could reflect a first low round of phosphorylation? Maybe by doing the same experiments with Ago1 to 3 the authors could see if just hAgo with miRNA are phosphorylated even if its low?

In Figure 2 is it possible to quantify remaining not bound mRNA to reinforce the idea that 14nt are the shortest length to establish base pairing?

The HDX-MS experiments show an important point: the target do not protect the EI site which was the simplest explanation. The results are also really interesting as it is suspected that free Argonaute are unstable but it’s one beautiful demonstration I saw in this direction.

Is it possible to test if phosphorylated hAgo2 loaded and with its target have a mRNA release decrease during a time course in presence of PP6, this would be demonstrated that when the phosphatase remove phosphorylation we observed a decreased of Koff.

---

## [Author Response]

Reviewer #1 (Recommendations for the authors):The results in figure 2 are quite interesting. Overall, you suggest that EI phosphorylation favour charge repulsion to release the miRNA/target association; however, this is especially true for some pairing degree with neither an extensive pairing nor a low pairing being the ideal. Can you discuss this better? Why do you think miRNA/target resolution depend on EI phosphorylation in some case more than others? Wouldn't all the cases need a negative charge to release the interaction? How do you think extensive pairing is resolved?

Figure 2 describes the ability of various targets to induce phosphorylation, rather than the ability of phosphorylation to promote target release. As we show in the figure, different targets promote phosphorylation to different extents. This would confound target release rate measurements, and therefore we were unable to compare the relative contributions of phosphorylation to target release between targets. However, because phosphorylation must occur to have any effect, targets that fail to induce robust phosphorylation would be expected to rely less on phosphorylation to aid in miRNA/target resolution and more on alternative mechanisms. Consistent with this, targets that failed to induce such phosphorylation either have less need for affinity-lowering (this is the case for seed-only targets, which have a lower affinity to begin with) or have alternative mechanisms to promote target release. In the latter category, targets with full complementarity can be sliced by hAgo2, which has been shown to enhance target release^1^. Targets with extensive 3’ pairing, but bulged central regions, can induce TDMD in vivo, which resolves the interaction through degradation of the RISC complex. We have added further discussion of this in the Discussion section (p 13, line 538-541).

“However, we found that extending 3’ pairing in the absence of central pairing also decreased EI phosphorylation (Figure 2b). Such targets can induce TDMD in vivo, which resolves the interaction through degradation of the RISC complex^63,64^, and therefore these targets would not need phosphorylation-promoted release to resolve RISC/target interactions.”

Figure 5, what happens if you only mutate S828 to Alanine, leaving the rest open? Can you prevent EI phosphorylation in vitro, and ideally in vivo, by mutating only S828? Even when it may be a little bit hard to test, it would be interesting to see whether S828 priming occurs in vivo.

We see no appreciable phosphorylation if S828 is mutated to alanine, even if all other sites are left open. We have added a figure showing this (Figure 5d). Although the in vivo experiments suggested are outside the scope of the current study, results from Golden et al., and Huberdeau et al., are consistent with S828 priming in vivo and we have added the following language to the Discussion section (p 12, line 476-487).

“This was further supported by their finding that S828A mutation, but not a single A mutation at any of the other phosphorylatable residues in the EI, eliminated detectable phosphorylation of hAgo2 in HCT116 cells lacking expression of ANKRD52, a regulatory subunit of PP6^2^. The central importance of S828 is also reflected in functional data from in vivo experiments. Huberdeau et al., found that mutating the corresponding serine in the *C. elegans* hAgo2 homolog ALG-1 (S992) to alanine led to severe developmental defects similar to those seen when mutating all EI phosphorylation sites to alanine^3^. Furthermore, Golden et. al reported that expression of hAgo2(S828A), but not wt hAgo2, was able to rescue repression of a miR-19 EGFP reporter in ANKRD52-deficient cells. Phosphorylation of S828 thus appears to serve as a key gatekeeper regulating further phosphorylation and its consequent functional effects, implying its regulation would be critical. We now show that this gatekeeper is directly regulated by target binding.”

Would it be possible to reanalyze data from ref 41 and see whether there is a bias toward the extent of miRNA/target pairing? Based on your hypothesis and discussion, one could expect more targets with 3' supplemental pairing in non-phosphorylated AGO2 mutants than in WT? If the reasoning of your discussion paragraph starting at Line 389 is correct, the in vivo results in this mutant should mimic better the in vitro data showing an increased affinity for more extensive pairing.

We thank the reviewer for their suggestion. We looked into the possibility of doing this analysis. Unfortunately, the processed eCLIP data in Golden et al., only yields target peaks (corresponding to hundreds or thousands of nucleotides) rather than precise miRNA binding sites. In that way, the eCLIP data are useful for comparisons at the mRNA level but not for miRNA/site-specific analyses. The eCLIP peaks typically encompass multiple predicted miRNA-binding sites, any of which could be bound. Additionally, the dataset lacks the identity of the corresponding bound miRNAs. We need this information for the requested analyses because members of the same miRNA family can differ in their extent of supplemental pairing. In order to directly compare wt- and 5XA-bound targets at the site level we would need miRNA/target site-specific data, such as CLEAR-CLIP results linking the Ago-bound miRNA to the target site to which it is bound (Moore et al., 2015)^4^.

The main open question here is the in vivo dynamics. When does the EI phosphorylation initiate and when the miRNA/target association is resolved? one would expect this is not immediate as the main RISC function is to suppress gene translation. Thus, quick phosphorylation and resolution wouldn't be ideal. Studying this aspect is beyond the scope of this paper and will need extensive analysis, but it would be nice if you discuss this point further.

We agree with the reviewer on this point and have added further discussion of this in the Discussion section to clarify our thinking. Additionally, we have now performed additional experiments showing that GW182 binding does not impact Ago phosphorylation, supporting our model that phosphorylation occurs after repression is established. These are described below in response to reviewer #2.

Out of personal curiosity, do you know how is EI Serine conservation in plant' AGOs (AGO1)? miRNA/target pairing is extensive in plants, so I would expect that this mechanism is not conserved? There is no need to discuss this on the manuscript.

We thank the reviewer for their interest. Serine-rich EI’s are conserved in plant Ago’s. However, the sequences and lengths of the EI vary. Interestingly, it was recently discovered that the EI’s of the *Arabidopsis thaliana* Ago proteins MAGO1 and MAGO2 are subject to phosphorylation in vivo, and are hypophosphorylated upon heat stress (Lee et al., 2021). We do not know at this juncture what the role is for EI phosphorylation in *Arabidopsis*. Although the responsible kinase was not identified in that study, another putative *Arabadopsis* CK1 target, the transcription factor DREB2A, gets stabilized upon heat stress due to decreased phosphorylation.

Phosphorylation normally occurs on DREB2A, leading to DREB2A degradation. But heat stress inhibits the phosphorylation, allowing DREB2A to accumulate and regulate stress response genes (Mizoi et al., 2019). Perhaps CK1α-mediated phosphorylation of hAgo could be reduced upon various stresses. However, this is only speculative. So far, although EI phosphorylation has been found to occur in a variety of organisms, the responsible kinase has only been identified in humans.

Reviewer #2 (Recommendations for the authors):I have a few suggestions for the authors:a) The use of purified components in an in vitro system is certainly a strength of this work, but it is also a limitation, since in vivo additional factors are known to interact with Argonaute (GW182 proteins as well as other effectors) and these interactions could significantly affect Ago phosphorylation.

We appreciate the reviewer’s suggestion and agree that it is important to keep in mind the effects additional factors, such as GW182, might have on Ago phosphorylation. Previous studies have found that EI phosphorylation does not impact hAgo2/GW182 interaction^2,3^. However, it remains a possibility that GW182 (or other recruited factors) could influence phosphorylation. Because GW182 plays a pivotal role in establishing target mRNA repression, binding directly to Ago and recruiting additional downstream effectors, we investigated the potential impact of GW182 binding on hAgo2 phosphorylation. We measured hAgo2 phosphorylation in the presence and absence of the GW182 Ago Binding Domain (GW ABD) (residues 455-860 of human TNRC6A)^5^ and found that GW ABD binding had no qualitatively detectable impact on endpoint phosphorylation of hAgo2 by CK1α. These results are compatible with our model that phosphorylation likely occurs after repression is established, thereby allowing effective repression without sequestering Ago longer than necessary. For these experiments, we had to use qualitative SDS-PAGE analysis of phosphorylation products instead of our quantitative scintillation counting method used in the experiments reported in our manuscript because we discovered that CK1α phosphorylates the GW ABD as well, confounding the measurement of total phosphorylation. Interestingly, although Ago phosphorylation was not impacted by GW ABD, GW ABD phosphorylation was greatly enhanced by the presence of hAgo2 (both wt and 5XA). As the GW ABD can bind up to 3 Ago molecules^5^, we suspect that, in these assays, hAgo2 is localizing CK1α to GW182, leading to its phosphorylation. However, we do not know where on GW this phosphorylation is occurring, what effects it might have on GW function, or whether it is physiologically relevant. Because of the issue of quantification, and since proper investigation of these questions requires further experiments that are outside the scope of the current paper, we have chosen not to include these preliminary results in the manuscript. However, we are providing them here to show that the immediate downstream factor associating with Ago does not interfere with Ago’s phosphorylation by CK1α, supporting the theory that Ago phosphorylation can occur after repression is established.

It would be perhaps useful to confirm the hyerarchical phosphorylation of Ago protein in a cell based experiment, for example by measuring phosphorylation of ectopically expressed Ago2 harboring various combinations of mutations in the 5 phosphorylation sites. The model predicts that the S828A and, to lesser extent, the S831A mutant should be resistant to phosphorylation, while mutants in the other three sites should still be at least partially phosphorylated.

This concern was also raised by reviewer #1. We now address this issue and added a new figure 5d. Please see our response to reviewer #1 above.

b) In the introduction (line 136), the authors mention ChIP-seq as the method used By Golden and colleagues to show reduced coverage of individual targets in the absence of phosphorylation. I think in the Golden paper the method used was eCLIP, not ChIP-seq.

We thank the reviewer for catching this typo! We were indeed referring to eCLIP experiments and we have corrected the text.

c) While this manuscript conclusively shows that S828 is the first residue to be phosphorylated by CK1 in a hierarchical fashion using full length Ago, a key role for S828 and S831 had been already suggested by Golden and colleagues using in vitro phosphorylation of various peptides corresponding to the EI region of Ago2 (Figure 4g). In fact, Golden et al., wrote: "Unphosphorylated peptide was a poor substrate for CSNK1A1 under these conditions, suggesting that initial phosphorylation of this region is facilitated by contextual features present in full-length AGO2. pS824 only weakly stimulated further phosphorylation. Prior phosphorylation of S828, however, robustly promoted phosphorylation of S831 (but not T830), while pS831 efficiently primed phosphorylation of S834. Taken together with our earlier data demonstrating a critical role for S828 in phosphorylation of AGO2 in cells (Figure 2e), these findings support a model whereby initial phosphorylation of S828, and potentially S824, stimulates efficient hierarchical phosphorylation of S831 followed by S834, rendering AGO2 incompetent for target binding until returned to an active state by ANKRD52-PPP6C phosphatase activity." It would be important to mention this in the discussion.

This is indeed a good point and in order to underscore the previous findings and clarify the added findings in this work, we have now added the following paragraphs in the Discussion section contextualizing our findings (p. 12, lines 461-487):

“The need for target-binding could explain findings from previously reported in-vitro phosphorylation assays testing CK1α’s ability to phosphorylated peptide substrates spanning hAgo2’s EI region^2^. Golden et al., reported that unphosphorylated EI peptide could not be phosphorylated by recombinant CK1α, leading them to propose that contextual features in the full-length hAgo2 were required in order to make S828 a suitable CK1α substrate. Our work shows that it is not only full-length hAgo2 that is required, but rather hAgo2 bound to a guide and a target RNA with seed and supplemental region pairing.

All the in vitro phosphorylation experiments that have been reported point to a central importance of S828. Our finding that S828 phosphorylation is required for subsequent hierarchical phosphorylation by CK1α in full-length hAgo2 (Figure 5) is consistent with previous experiments using phosphorylated peptide substrates^2^. Golden et al., showed that peptides phosphorylated at S828 were able to be phosphorylated at S831 and those phosphorylated at S831 were able to be phosphorylated at S834. They suggested that pS828 primes for hierarchical phosphorylation of the EI at S831 and S834, consistent with our experiments with full-length hAgo2. This was further supported by their finding that S828A mutation, but not a single A mutation at any of the other phosphorylatable residues in the EI, eliminated detectable phosphorylation of hAgo2 in HCT116 cells lacking expression of ANKRD52, a regulatory subunit of PP6^2^. The central importance of S828 is also reflected in functional data from in vivo experiments. Huberdeau et al. found that mutating the corresponding serine in the *C. elegans* hAgo2 homolog ALG-1 (S992) to alanine led to severe developmental defects similar to those seen when mutating all EI phosphorylation sites to alanine^3^. Furthermore, Golden et al., reported that expression of hAgo2(S828A), but not wt hAgo2, was able to rescue repression of a miR-19 EGFP reporter in ANKRD52-deficient cells. Phosphorylation of S828 thus appears to serve as a key gatekeeper regulating further phosphorylation and its consequent functional effects, implying its regulation would be critical. We now show that this gatekeeper is directly regulated by target-binding.”

Reviewer #3 (Recommendations for the authors):Line 165 Please explain why you choose miR-200b

We have added the following (p. 5, lines 170-174):

“This guide/target combination was chosen because the target’s genomic location was identified as bound to 5XA but not wild-type hAgo2 in previous eCLIP experiments^2^, suggesting it was sensitive to EI phosphorylation in vivo. It was also chosen because of its functional relevance in helping regulate epithelial to mesenchymal transitions^6^.”

Figure 1c why mutating all residue at the same time and not testing simple mutation and combination? this experiment is done in figure 5 and could maybe be positioned in figure 1.

We understand the logic behind this suggestion, however, the aim of Figure 1c is to show that target binding triggers CK1α-mediated phosphorylation and that this phosphorylation is occurring in the eukaryotic insertion, hence the use of the 5XA mutant in this situation. As we do not discuss the precise locations of the phosphorylations until later in the paper, we have left the experiments showing the effects of single A mutations and combinations in Figure 5. We have also added a figure showing the effects of each single A mutation (Figure 5d).

Is hAgo2 plus guide phosphorylation in the expected background or it could reflect a first low round of phosphorylation? Maybe by doing the same experiments with Ago1 to 3 the authors could see if just hAgo with miRNA are phosphorylated even if its low?

The levels of phosphorylation in guide-bound Ago1/2/3 are within the expected background accounting for artifacts of the experimental system (Figure 1 —figure supplement 1). These experiments are performed to measure endpoint or near-endpoint phosphorylation, after 90 minutes, so multiple rounds of phosphorylation are possible. However, the phosphorylation with guide-bound alone does not exceed the expected background even with the long incubation time.

In Figure 2 is it possible to quantify remaining not bound mRNA to reinforce the idea that 14nt are the shortest length to establish base pairing?

We don’t believe one needs 14-nt of target to establish base-pairing per se. As we show in Figure 2 —figure supplement 1, the affinity for a target of 13-nt in length is very similar to that of 14-nt long. However, the differences in the amounts of phosphorylation triggered by these targets are striking. Therefore, the requirements for phosphorylation shown in Figure 2 go beyond simple base pairing.

The HDX-MS experiments show an important point: the target do not protect the EI site which was the simplest explanation. The results are also really interesting as it is suspected that free Argonaute are unstable but it’s one beautiful demonstration I saw in this direction.Is it possible to test if phosphorylated hAgo2 loaded and with its target have a mRNA release decrease during a time course in presence of PP6, this would be demonstrated that when the phosphatase remove phosphorylation we observed a decreased of Koff.

We thank the reviewer for this suggestion. The proposed experiments would be difficult to interpret conclusively, unfortunately, because the timescales of dephosphorylation and target release would be confounding one another. Furthermore, in vivo, PP6 functions as a heterotrimer and although 2 of the subunits were identified in Golden et al.,’s CRISPR screen (ANKRD52 and PPP6C), the identity or identities of the third subunit involved remain unknown.